# Triaxial Load Cell for Ergonomic Risk Assessment: A Study Case of Applied Force of Thumb

**Mario Acosta-Flores** [1] , **Martha Roselia Contreras-Valenzuela** [1,*] , **J. Guadalupe Velásquez-Aguilar** [1] , **Francisco Cuenca-Jiménez** [2] **and Marta Lilia Eraña-Díaz** [1,*]

1 Faculty of Chemical Sciences and Engineering, Autonomous University of Morelos State (UAEM), Avenida Universidad 1001, Colonia Chamilpa, Cuernavaca CP 62209, Mexico; mario.acosta@uaem.mx (M.A.-F.); jgpeva@uaem.mx (J.G.V.-A.)

2 Department of Mechanical Engineering, National Autonomous University of Mexico (UNAM), Coyoacán, Mexico City CP 04510, Mexico; francisco.cuenca@ingenieria.unam.edu

* Correspondence: marthacv@uaem.mx (M.R.C.-V.); merana@uaem.mx (M.L.E.-D.);
Tel.: +52-7772493629 (M.R.C.-V.); +52-7773280087 (M.L.E.-D.)

**Abstract:** To assess the ergonomic risk level in work systems involving tasks performed with hands or fingers, it is necessary to know the exerted triaxial forces. To address this need, a prototype of a triaxial load cell based on principles of linear elasticity theory and mechanical problems of torsion, bending and axial load is presented. This work includes an analytical strain model for each instrumented point and its solution regarding the applied force to a triaxial load cell. The proposed load cell was calibrated and validated by performing different static experimental tests. As a case study, the applied force in three directions while the thumb activates a cigarette lighter was measured. Triaxial forces and resultant forces were obtained and compared with the parameter of 10 N established by the ergonomic standards as reference values for pressing down with the thumb, finding that the applied forces in eight tests were 23.73 N, 43.51 N, 12.69 N, 14.50 N 20.35 N, 21.67 N, 39.74 N and 46.02 N, exceeding the reference values and establishing a direct relationship with Quervain syndrome. In conclusion, the developed load cell is a valid and reliable alternative to measure many forces that cannot be obtained with commercial devices, allowing the level of ergonomic risk to be determined with great precision.

**Keywords:** triaxial load cell; mechanical design; ergonomics; risk assessment; biomechanics

## 1. Introduction

Measuring the applied force by fingers during task performance in real time can be challenging due to various factors. The low loads handled at high frequencies make it difficult to obtain accurate measurements. Additionally, rapid movements and sweat can cause the sensors placed on the skin to become loose or fall off. To overcome these issues, this paper proposes a prototype of a triaxial load cell. This device will provide more reliable measurements of the applied thumb force, ensuring better overall accuracy. The cell has been tested in a case study to validate its performance in monitoring the force exerted by the right thumb during a quality control task that involves testing disposable classic cigarette lighters to modulate the size of the flame.

To determine if work tasks are causing work-related illnesses, ergonomic assessments must be conducted to monitor the interaction between humans and tools or products. When assessing tasks performed by hand, measuring the force exerted (applied force AF) is essential to ensure that the force falls within established safety standards [1]. If the AF magnitude is unsafe, it will be necessary to implement preventive actions [2]. Fine motor movements of the hand, particularly in the wrists or fingers, can be evaluated through AF measurements [3]. The muscle strength during isometric contraction remains relatively constant, with minimal variation in muscle length [4]. However, small loads generate

AF during repetitive movements at a high frequency [2]. This means prolonged exposure to AF can result in high muscular loads, which is considered an ergonomic hazard (EH) [5–8]. Work-related musculoskeletal disorders (WMSDs) are injuries in muscles, tendons, joints, and nerves due to EHs. Workers' most prevalent hand diseases are carpal tunnel syndrome, tendinitis, inflammation and repetitive motion injuries. These conditions cause pain and reduced wrist or finger mobility, negatively impacting employees' well-being [9]. Roberts et al. proposed a standardised technique for measuring grip strength in this context [10]. The method is squeezing a handgrip dynamometer for 3 s with maximum voluntary effort (MVE). It is a common technique used to evaluate the effects of AF on injuries and to monitor the progress of rehabilitating injured hands. It is important to note that this technique is used to follow up on the recovery of an injury and to monitor rehabilitation progress rather than as a tool to prevent ergonomic risks. Thus, the MVE method is not applicable for real-time ergonomic risk assessment in production areas. The mechanical problem of measuring AF developed by fingers requires knowledge of geometry, the properties of materials, and the conditions for their definition (forces, displacements, speeds, accelerations, temperatures, etc.) [11]. When forces exerted by hands and fingers are measured in real time, estimating the magnitude of triaxial forces is a significant challenge, i.e., force components should be monitored in three directions (x-, y-, and z-axis) during work movements, such as flexion, extension, radial deviation, ulnar deviation and torsion. The measurement technique must not interfere with work development and should not be invasive.

A load cell transducer converts force into a measurable electrical output [12]. A load cell is a mechanical component that is linear elastic. When a force is applied, the load cell undergoes deformation, generating internal stresses and strains. These are detected by electrical extensometers, which convert them into an electrical signal. By using a calibration factor to this signal, the value of the applied force can be determined accurately [11,13,14]. An extensometer, also known as a strain gauge, is a type of sensor that can measure strain by detecting a change in electrical resistance resulting from tension or compression applied to the load cell. The strain parameter can either be positive, indicating tension, or negative, indicating compression. A Wheatstone bridge circuit is used to measure electrical resistances and calculate the charge [15]. It contains strain gauges, which improve the output signal resolution, compensate for temperature changes and eliminate specific mechanical effects, such as deflections due to asymmetry [11]. The output signal value is determined by a combination of the following factors: geometry of the mechanical elements, the elastic properties, the sensors' location and the AF magnitude on the cell.

In the market, the main commercial devices were designed to measure the exerted force by the hand using the MVE method [16–18]. The choice of measurement technology can significantly impact the accuracy of force measurements, with certain technologies being more suitable for specific types of force measurement than others. In the literature, there are two kinds of devices: commercial devices and devices developed to resolve specific problems for investigation purposes. Commercial devices are used in rehabilitation procedures, e.g., Bretz et al. [19] introduced an intriguing device known as the Dyna-8, a portable measurement system enabling users to gather specific numerical data and diagrams on hand and finger forces. This system records the maximum force value through a microcontroller and displays it digitally on an electronic unit. In [20], a finger force sensor adapter is proposed, which measures the maximum grip force control and force tracking task; it monitors and quantifies the patient's process during therapy. Each of these measure effort and torque. In contrast, a force glove system, such as the Force Sensitive Application (FSA) system, the Measure Grip Forces, and the Hand Force Measurement System [21–23] all directly measure finger forces exerted on an object; they use a sensitive resistor sensor attached to the palmar side of the hand and have high precision in their measurement. However, they limit the natural movements of the hand and fingers in repetitive work, change task conditions and affect productivity; this aspect limits their use when workers perform a high-frequency task with small loads. Therefore, none of them can be used in a

production process to measure AF in real time, as required in ergonomic and biomechanical risk assessments. The Body Pressure Measurement System (BPMS) [23] measures pressure distribution throughout the body and support surfaces. Unfortunately, it does not measure triaxial forces; its measurements are perpendicular and unidirectional to the plane.

On the other hand, an example of the devices developed to resolve specific problems is the 3D force sensor for biomechanical applications proposed by Brookhuis et al. [24], which aims to integrate sensors in a glove to determine the complete mechanical interaction between the human hand and its environment. The sensor measures the normal force and two perpendicular moments by dividing the read-out into four quadrants. Martelli et al. [25] proposed an optical fibre sensor for measuring joint angles. Unfortunately, the sensor is invasive; it measures intra-articular pressure. Mandy et al. [22] measured hand/hand grip forces in one-arm drive wheelchairs using the grip force. Dahlqvist et al. [26] compared hand grip and resisted wrist extension (using MVC) in terms of amplitude and reproducibility and examined the effect of electrode positioning using electromyography through Ag/AgCl electrodes, which are designed for electrochemistry, stimulation, precision bioelectric recording and electrophysiology; however, this procedure is far from providing accurate information regarding the applied force since it only measures unidirectional muscular load. In a related context, Pinder et al. [27] noted a common trend in studies focusing on whole-body manual strength, which often emphasize forces exerted in either the sagittal plane or solely in a vertical direction.

The triaxial load cell (TLC) introduced in this study serves as a transducer that transforms force into an electrical output signal, facilitating real-time linear measurement and correlation with the applied force. A notable technological edge over conventional force measuring devices is that the TLC can be integrated onto or within the tool or product being handled by workers during task execution, eliminating the need for placement on the skin or hand. Thus, the prototype load cell proposed in this work differs from other devices in its use and can be integrated into ergonomics, biomechanics and human–machine interaction system applications. A comparison of characteristics between the current device and the TLC proposed in this paper is shown in Table 1. As can be seen, the novelty lies in measuring the applied force exerted by hands or fingers in real time, with a device implemented in the product or tool, with a non-invasive measuring technique.

**Table 1.** Comparison of characteristics between devices that measure the applied force from the wrist.

| Author | It Measures in Real Time | It Is Non-Invasive | It Can Be Implemented in Tools | It Can Be Used during Task Development | It Measures the Triaxial Force |
|---|---|---|---|---|---|
| H. C. Roberts et al. [11] | ✓ | ✓ | | | |
| K. Bretz, et al. [20] | ✓ | ✓ | | | |
| Tekscan, Tactile Grip Measurement System [21] | ✓ | ✓ | | | |
| K. Jung et al. [22] | ✓ | ✓ | | | |
| A. Mandy, et al. [23] | ✓ | ✓ | | | |
| Tekscan, Body Pressure Measurement System (BPMS) [24] | ✓ | ✓ | ✓ | | |
| R. Brookhuis, et al. [25] | ✓ | ✓ | ✓ | ✓ | |
| C. Martelli, et al. [26] | ✓ | | ✓ | | |
| C. Dahlqvist, et al. [27] | ✓ | | ✓ | ✓ | |
| D. Pinder, et al. [28] | ✓ | ✓ | ✓ | | ✓ |
| Three axial load cell proposed in this paper | ✓ | ✓ | ✓ | ✓ | ✓ |

The rest of this paper is organised as follows: The Materials and Methods are described in detail in Section 2. The Results of this study are presented in Section 3. The Discussion and Conclusion of the investigation are presented in Sections 4 and 5, respectively.

## 2. Materials and Methods

As a problem statement, the most common injuries and diseases in hands are usually caused by repetitive efforts over long periods associated with automated and semi-automated work systems. Five hand anatomical movements are present in work tasks: hand and finger flexion (e.g., grip), extension (when the hand is pulled back), radial deviation (when the wrist is moved towards the midline in adduction) and ulnar flexion (when the wrist is carried away in abduction). All of them are directly associated with WMSDs. Therefore, the prototype of the triaxial load cell (at the laboratory level) must measure the components of the AF during task performance. This implies measuring the magnitude and force components in three directions (x, y, z) during work movements, considering the speed and repetition per unit of time, without intervening in work performance or changing the movements of the hand. Consequently, the most critical requirement is a load cell with a geometry adapted to the tool manipulated by the worker.

### 2.1. Study Case

The cigarette lighter manufacturing process requires a 100% function test and flame adjustment. The manually performed task consists of activating the cigarette lighter with the right thumb and adjusting the height of the flame with the left hand, with a holding time of 2 to 3 s, See Figure 1. (To consult the video, access the following link https://youtu.be/pWkFmd81ssw (accessed on 30 April 2024)). Workers in the area have developed two types of musculoskeletal disorders: carpal tunnel syndrome and Quervain syndrome in the right hand. To define if the AF was the cause of the WMSDs development, it is necessary to know if the applied force when activating the lighter exceeds the established safety standards given in ISO11228-3 [2] as follows: reference values when thumb pressing down the micro spark wheel (control actuator) of the cigarette lighter with an MVE of 100 N and a maximal permissible force of between 10 N and 25 N. Therefore, it is necessary to measure the force exerted by the thumb.

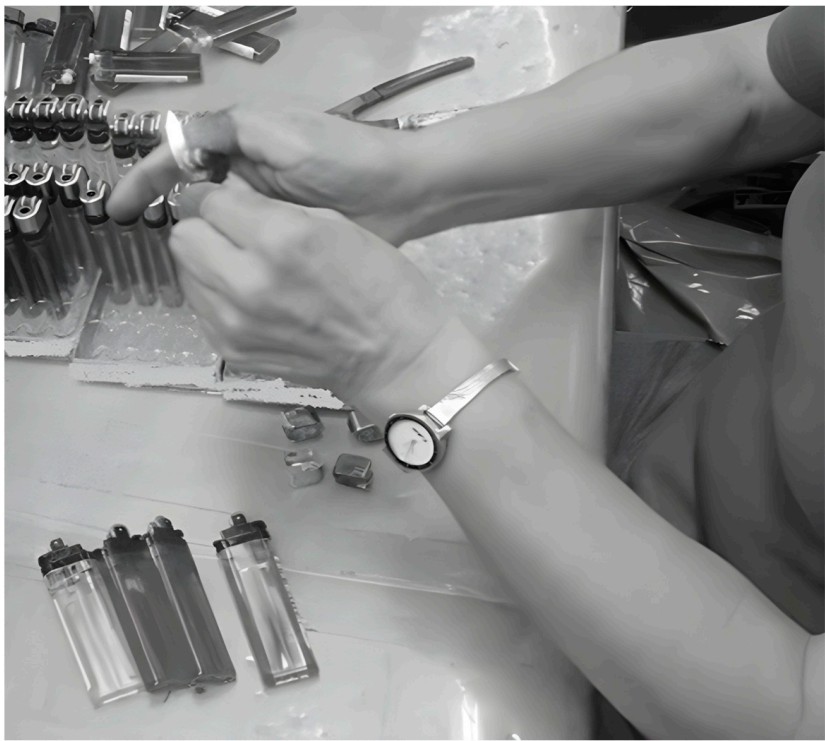

**Figure 1.** The task is to activate a cigarette lighter with the right thumb and adjust the flame height with the left hand.

### 2.2. Method of Design

Measurements were taken to determine the AF (applied force) during the thumb's activation of the cigarette lighter in three directions. As this force is triaxial (with components in the x-, y- and z-axis), the design was based on solving simple mechanical problems for symmetric elements. Considerations included axial load, beam bending and torsion problems in rectangular cross-section bars for an exact solution. The triaxial load cell was experimentally instrumented with electrical extensometers for testing and measuring. The development process for the triaxial load cell prototype involved the following steps:

1.  Design of the cell mechanical structure;
2.  Development of the analytical model;
3.  Construction of load cells;
4.  Load cell configuration and instrumentation;
5.  Testing and Calibration model;
6.  Evaluation of Efficiency Degree;
7.  Experimental Test in Ergonomic Assessment.

### 2.3. Design of the Mechanical Structure of the Cell

The geometric dimensions must be determined based on the maximum value of the triaxial force to be measured plus a value as a safety factor. The design must be implemented in any tool manipulated in any workstation. Therefore, the load cell design was proposed with an "L" shape, as described in Figure 2, built with the union of two bars. Their dimensions are established in Table 2.

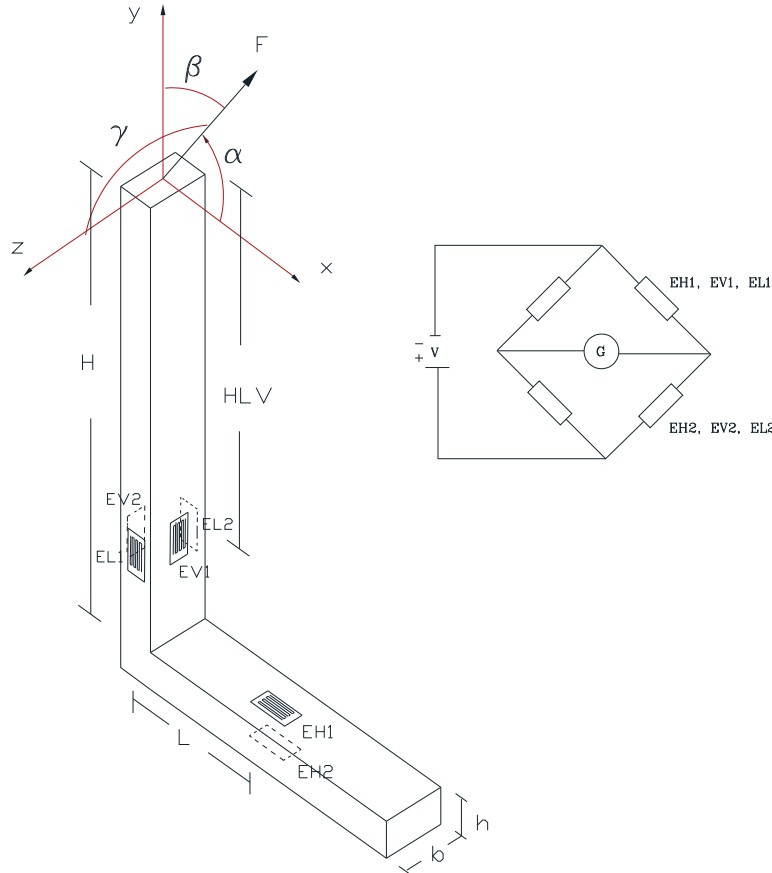

**Figure 2.** Triaxial load cell design, load cell in L shape and its instrumentation. H is the vertical element length, HLV is the distance from the application point to strain gage EVP, L is the distance from the application point to strain gage EHP, b is the transversal high, h is the horizontal length and Fx, Fy and Fz denotes the force in the x-, y- and z-axes, and α, β and γ are the forces angles.

**Table 2.** Load cell dimensions.

| Dimension | (m) |
|---|---|
| b | 0.016 |
| h | 0.01 |
| L | 0.0327 |
| H | 0.1347 |
| HLV | 0.0978 |
| LT | 0.075 |

The proposed material is aluminium with a Young's modulus of 70 GPa, which provides a higher level of deformations and a more excellent cell resolution. The mechanical evaluation of the cell structure was carried out considering the worst load condition, for which the location of points had the most relevant state of stress. The x-axis direction was considered the design condition of the resultant force. A failure criterion as the maximum distortion energy was determined using Equation (1).

$$N = \frac{\sigma_y}{\sigma'} \tag{1}$$

where N is a factor design, $\sigma_y$ is the material's yield stress and $\sigma'$ is the Von Mises stress.

For a maximum $F_x$ of 40 N, an aluminium material with an elastic limit of 50 Mpa and a Von Mises stress at the points marked as critical, a design factor of N = 2.45 was obtained. The cell design should be changed based on the dimensions and cross-section for a higher load capacity.

*2.4. Instrumentation Points*

The positioning of the points to be instrumented must consider the Saint Venant Principle. Thus, the points should be located at a minimum distance of the largest dimension value of the cross-section sides concerning the load application points or concerning zones where there is a geometric change; this is important because this ensures that the output strains provided by the strain gauges are reliable and represent the actual strain state of a point represented by a linear mechanical model of combined stresses, integrated by the analytical models' bending, axial load and torsion. The location of extensometers on the cell was defined using a Wheatstone half-bridge connection, as depicted in Figure 2.

- Strain gages ($\varepsilon_{V1}$, $\varepsilon_{V2}$, $\varepsilon_{H1}$ and $\varepsilon_{H2}$) measure the force in the x-axis direction.
- Strain gages ($\varepsilon_{L1}$, $\varepsilon_{L2}$ $\varepsilon_{V1}$, $\varepsilon_{V2}$ $\varepsilon_{H1}$ and $\varepsilon_{H2}$) measure the force in the y-axis direction.
- Strain gages ($\varepsilon_{L1}$ and $\varepsilon_{L2}$) measure the force in the z-axis direction.

The arrangement of strain gages positioned on the cell neutral axis eliminates the signals from the noise produced by changes in room temperature and moments generated by torsion effects. The pair of extensometers placed on the bottom of the cell ($\varepsilon_{H1}$ and $\varepsilon_{H2}$) monitor the deformation caused by the components in the x- and y-axes but eliminate the signals generated by the force component in the z-axis. Integration of the load cell into the structure or machine where the force is to be measured can be performed by welding, glueing or screwing. If the integration is by welding, the instrumentation must be carried out after this process because the high temperature caused by the process could damage the strain gages; however, if the union is carried out using glue or screws, the instrumentation could be carried out earlier.

*2.5. Analytical Model*

The mechanical analytical model of the load cell was based on the theory of linear elasticity. The following aspects were assumed: the model is linear, i.e., the working range considers infinitesimal strains under an elastic linear behaviour, so the load cell was manufactured using homogeneous and isotropic material. Additionally, the model combines stress problems (axial load problem, bending problem and torsion problem).

Thus, the state of stress and strain in the instrumented points of the load cell was analysed based on the analytical models existing in the literature [11,28,29].

The analytic solution in strain terms is a function of each instrumented point on the load cell. The input equations were determined specifically for the mechanical model proposed in the function of strain, axial load, moments, torsion and extensometer position. The analytical equations system was defined as a function of instrumented points and the geometric structure of the cell. The resulting equations in terms of strains at instrumented points and in the direction of the measurement of strain gages are:

$$\varepsilon_{H1} = \frac{1}{E}\left[ -\frac{F_x}{bh} - \frac{6F_x H}{bh^2} - \frac{6F_y L}{bh^2} \right] \tag{2}$$

$$\varepsilon_{V1} = \frac{1}{E}\left[ \frac{F_y}{bh} - \frac{6F_x HLV}{bh^2} \right] \tag{3}$$

$$\varepsilon_{L1} = \frac{1}{E}\left[ \frac{F_y}{bh} - \frac{6F_z HLV}{b^2 h} \right] \tag{4}$$

$$\varepsilon_{H2} = \frac{1}{E}\left[ -\frac{F_x}{bh} + \frac{6F_x H}{bh^2} + \frac{6F_y L}{bh^2} \right] \tag{5}$$

$$\varepsilon_{V2} = \frac{1}{E}\left[ \frac{F_y}{bh} + \frac{6F_x HLV}{bh^2} \right] \tag{6}$$

$$\varepsilon_{L2} = \frac{1}{E}\left[ \frac{F_y}{bh} + \frac{6F_z HLV}{b^2 h} \right] \tag{7}$$

where

$\varepsilon_{H's}$, $\varepsilon_{V's}$ and $\varepsilon_{L's}$ denote horizontal strain, vertical strain and lateral strain, respectively; E is the Young's module of the material load cell, H is the vertical element length, HLV is the distance from the application point to strain gage EVP, L is the distance from the application point to strain gage EHP, b is the 16 mm transversal high, h is the 10 mm horizontal length and $F_x$, $F_y$ and $F_z$ denote the force in the x-, y- and z-axes in that order.

The resulting system of linear equations (see Equations (2)–(7)) from the mechanical analysis was solved with the help of Wolfram Mathematica; consult Appendix C to see the program. The resulting solution was $F_x$, $F_y$, $F_z$ and FR (Equations (8)–(14)). Moreover, the Wheatstone bridge half connection was used to eliminate the effects of the temperature and mechanical conditions required in the ideal mechanical problems (in the axial load flexion and torsion models). Because all equation solutions are in the function of Young's module (E), the calibration constant (K) replaced it; this procedure allows the cell calibration in the three directions. K represents the three-dimensional calibration coefficient (from cell calibration experimental data in Pa, which will be determined in the following section).

$$F_x = \frac{K\varepsilon_V bh^2}{12HLV} \tag{8}$$

$$F_y = \frac{K\varepsilon_V Hbh^2 - K\varepsilon_H HLVbh^2}{12HLVL} \tag{9}$$

$$F_z = \frac{K\varepsilon_L b^2 h}{12HLV} \tag{10}$$

with resultant force

$$FR = \sqrt{F_x{}^2 + F_y{}^2 + F_z{}^2} \tag{11}$$

and

$$\alpha = \cos^{-1}\frac{F_x}{FR} \tag{12}$$

$$\beta = \cos^{-1}\frac{F_y}{FR} \tag{13}$$

$$\gamma = \cos^{-1}\frac{F_z}{FR} \tag{14}$$

where α, β and γ are the angles the resultant force adopts concerning the x-, y- and z-axes.

### 2.6. Testing and Calibration Model

The calibration of each one of the cells shall be carried out individually, determining the value of K (which replaces Young's modulus) for different applied loads. Equations (15)–(17) were used to determine the constant of each cell $K_H$, $K_L$ and $K_V$ for the horizontal, lateral and vertical cells, respectively.

$$K_H = \frac{6PL}{bh^2\varepsilon_H} \tag{15}$$

$$K_L = \frac{6PHLV}{hb^2\varepsilon_L} \tag{16}$$

$$K_H = \frac{6PHLV}{bh^2\varepsilon_V} \tag{17}$$

It is essential to consider that the obtained K factor is valid specifically for the dimensions determined for H, HLV and L in this document. If the dimensions are altered, it is strongly advised to recalibrate the load cell using the equations provided above.

### 2.7. Building of the Triaxial Load Cell Prototype

The design of the triaxial load cell was tailored to meet the requirements of a risk prevention device capable of measuring triaxial forces generated when the thumb activates a cigarette lighter. The design must consider the requirement of being implemented in any tool manipulated in a workstation. The prototype was built using two aluminium bars fixed with epoxy resin and two screws with a diameter of 4 mm and a length of 16 mm. The mechanical body of the cell has an "L" shape and a rectangular cross-section. An acrylic plate was adapted as the base of the horizontal bar (in a real process, the TLC can be adjusted in a hidden way on or inside any workstation surface, tool or product); on the plate, a pad with cables was fixed to transmit the signal from the extensometers to the Wheatstone bridge. To obtain experimental data, the cell was instrumented using strain gages. A strain indicator and recorder were used for dynamic and static monitoring; moreover, for data acquisition, a Strain Gage Vishay (USA): CEA-13-240UZ-120 was implemented.

The calibration was carried out individually for each internal cell to determine the value of *K* for different applied loads. Experimental tests were developed to determine K through a simulation of AF using standardised weights (W) ranging from 0.1 kg to 2.25 kg, as is depicted in Figure 3. The results and analysis are shown in Appendices A and B, respectively. With the resulting deformations and the test model, the average calibration constant K was obtained for the triaxial load cell established in Equations (15) to (17). The results were as follows: KH = 70,761,838,235 Pa, KL = 70,892,578,125 Pa and KV = 71,222,132,813. The average represents the three-dimensional calibration coefficient KAV = 70,958,849,725. The complete test results are in Tables A1–A3 of Appendix A.

The application of forces can be performed either by applying dead weights or by using known forces with the support of another load cell or, if possible, with the help of a universal machine. The KAV was the value used in Equations (8)–(10) to determine the resulting triaxial force and its components (FR, Fx, Fy and Fz). For each test, known forces (W) were applied in different directions using α, β and γ angles. The results from test 1 are shown in Table 3. Applying Equations (8)–(11), the force components (Fx, Fy and Fz) and the resulting force FR were determined, and the complete experimental data from four tests with 15 known weights are provided in Tables A4–A6 of Appendix B.

The deviation between the known force (W) and the applied force (AF) measured with the triaxial load cell was included as a validation procedure, like the percentage of deviation. As is observed in Table 3 and Appendix B's results, the variation rate was small, with a mean of 2.02% and standard deviation, SD, of 1.13. Thus, the triaxial load cell was

ready to be tested as an ergonomic assessment tool. For this purpose, a cigarette lighter activation was used as a case study. The mechanical element was embedded over an acrylic base with a well-centred lighter at the top, see Figure 4. This layout allowed testing of the thumb movements required in the case study. The cigarette lighter was interchangeable; thus, changing the cigarette lighter for other tools, such as a button or a different handle geometry, was possible, resulting in different task simulations.

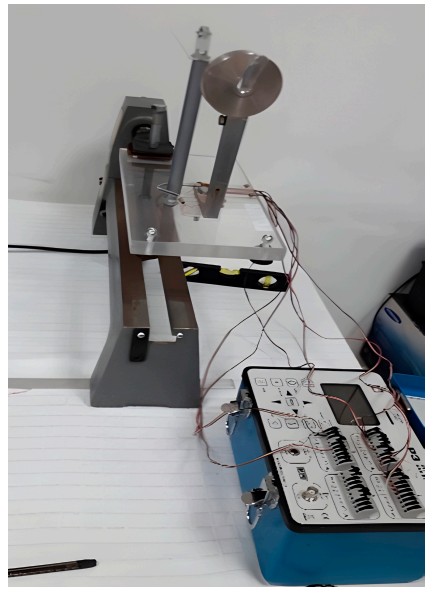 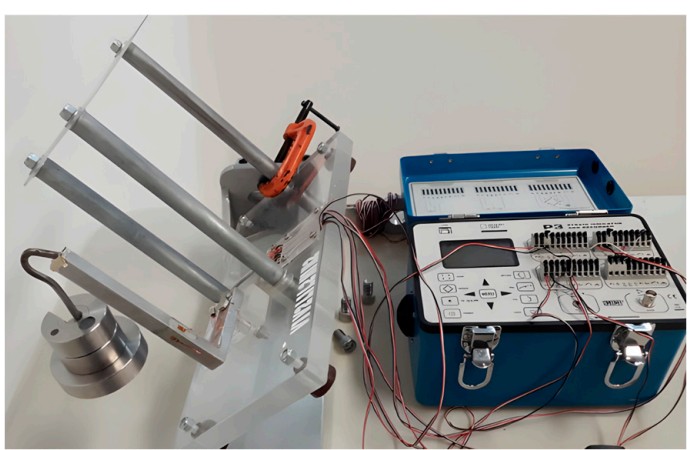

**Figure 3.** Calibration and testing of the TLC; the thumb AF was simulated using standardised test weights ranging from 0.1 kg to 2 kg. Different $\alpha$, $\beta$ and $\gamma$ angles (which simulate the hand position) determine the three-dimensional calibration coefficient KAV. The data were acquired using a Strain Gage Vishay (USA): CEA-13-240UZ-120.

**Table 3.** Experimental evaluation for an applied force. Test 1 with $\alpha$ = 0° and $\beta$ = 43°.

| W (N) | $\varepsilon_H$ (με) | $\varepsilon_V$ (με) | $\varepsilon_L$ (με) | K (Pa) | Fx (N) | Fy (N) | Fz (N) | AF (N) | % of Deviation |
|---|---|---|---|---|---|---|---|---|---|
| 0.981 | −2 | 0 | 4.5 | 70,958,849,724 | 0.000 | 0.652 | −0.711 | 0.965 | 1.62 |
| 1.962 | −5.5 | −1 | 9.5 | 70,958,849,724 | 0.099 | 1.335 | −1.502 | 2.012 | 2.49 |
| 2.4525 | −6.5 | −1 | 12 | 70,958,849,724 | 0.099 | 1.662 | −1.897 | 2.524 | 2.83 |
| 3.4335 | −10 | −2 | 17 | 70,958,849,724 | 0.198 | 2.345 | −2.688 | 3.572 | 3.88 |
| 4.4145 | −12 | −2 | 22 | 70,958,849,724 | 0.198 | 2.997 | −3.478 | 4.596 | 3.94 |
| 4.905 | −12.5 | −2 | 23 | 70,958,849,724 | 0.198 | 3.160 | −3.636 | 4.822 | 1.73 |
| 5.886 | −15 | −2 | 27 | 70,958,849,724 | 0.198 | 3.976 | −4.269 | 5.837 | 0.84 |
| 8.3385 | −21 | −3 | 39 | 70,958,849,724 | 0.296 | 5.474 | −6.166 | 8.251 | 1.06 |
| 9.81 | −25 | −2 | 45 | 70,958,849,724 | 0.198 | 7.238 | −7.114 | 10.151 | 3.36 |
| 10.791 | −23 | −4 | 50 | 70,958,849,724 | 0.395 | 7.299 | −7.905 | 10.767 | 0.23 |
| 12.2625 | −31 | −5 | 57 | 70,958,849,724 | 0.494 | 7.819 | −9.012 | 11.941 | 2.69 |
| 14.715 | −38 | −6 | 69 | 70,958,849,724 | 0.593 | 9.644 | −10.909 | 14.572 | 0.98 |
| 17.1675 | −44 | −7 | 80 | 70,958,849,724 | 0.692 | 11.142 | −12.648 | 16.870 | 1.76 |
| 19.62 | −50.5 | −8 | 92 | 70,958,849,724 | 0.790 | 12.804 | −14.545 | 19.394 | 1.17 |
| 22.0725 | −56.5 | −9 | 103 | 70,958,849,724 | 0.889 | 14.302 | −16.284 | 21.691 | 1.76 |
| | | | | | | | Mean | | 2.02 |
| | | | | | | | SD | | 1.13 |

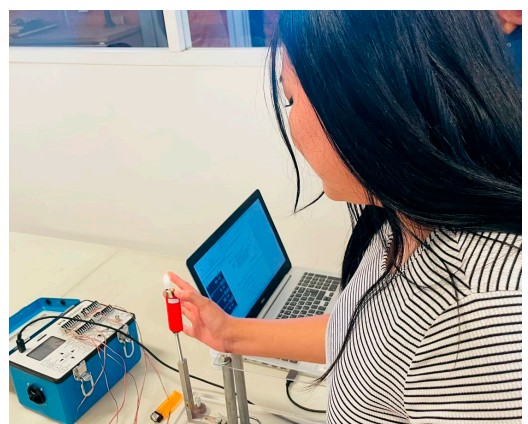
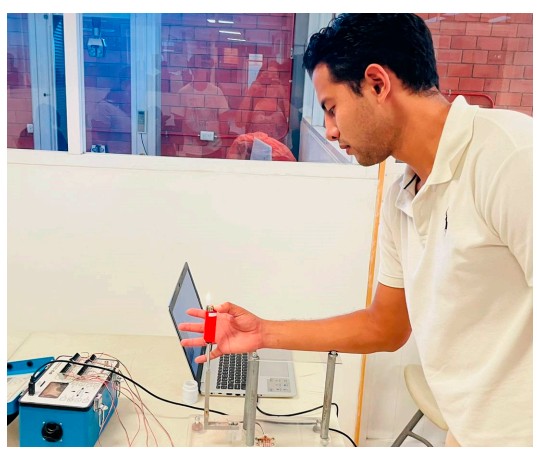

**Figure 4.** Biomechanical evaluation at the laboratory level through measuring the applied force of the thumb in a task to activate a cigarette. Each person performed five repetitions, turning on the lighter and maintaining pressure with the thumb for 3 s. The force provided by the load cell is only when the thumb presses down the micro spark wheel. The anthropometric data are presented in Appendix E.

## 3. Results

*Measuring the Applied Force of Thumb during the Activation of a Cigarette Lighter*

The triaxial load cell was designed as an ergonomic and biomechanical assessment tool. Therefore, the results obtained from monitoring the AF developed by the thumb (when the cigarette lighter was activated; see Figures 1 and 4) were compared with the parameters established by [2] and [30] to consider a possible risk factor. In the analytical model, Equations (8)–(11), the following values were considered: b = 0.016 m, h = 0.01 m, L = 0.029 m, H = 0.277 m and HLV = 0.2451 m. The objective of the test was to identify whether the thumb force applied when lighting a lighter did not exceed the established safety standards (reference values when the thumb pressing down the micro spark wheel considered as a control actuator) with a low risk of developing WMSDs with the force magnitude less than 10 N, a medium risk between 10 N and 25 N and a maximal permissible force of 25 N; therefore, the force magnitude greater than 25 N represents a high risk of developing WMSDs. Each person performed five repetitions, turning on the lighter and maintaining pressure with the thumb for 3 s. The force provided by the load cell only occurs when the thumb is pressing down the micro spark wheel. Figure 4 shows the general operation and a representation of the operation carried out in the study. The results are shown in Table 4 and in the tables in Appendix D. The task must be considered very demanding and assessed as high risk to develop WMSDs. Table 4 presents the eight maximum measured forces. The AF average for females was 24.12 N, and for males it was 31.42 N. The forces on the *x*-axis and the forces on the *z*-axis did not represent an EH because the magnitudes were below 10 N. However, the forces in the *y*-axis exceeded the maximal permissible value of 25 N in 38% of the cases, representing an ergonomic risk factor for developing WMSDs. The movement in the *y*-axis represents the pressing down force of the thumb, affecting the finger flexor/extensor tendons and synovia. Consequently, when the applied force surpasses 10 N in a single motion, the task is leading to the development of De Quervain's tenosynovitis (Quervain syndrome) among female workers. Additionally, there is a potential risk of this exerted force causing Carpal tunnel syndrome.

In Figure 5 and Appendix D, the resulting data in $\mu\varepsilon$ of $\varepsilon_H$, $\varepsilon_V$ and $\varepsilon_L$ from monitoring in the real-time activity are presented. The Equations (8)–(10) can convert the data into forces. It is important to note that, for this case study, the maximum test values vary between the range of 30 N and 50 N.

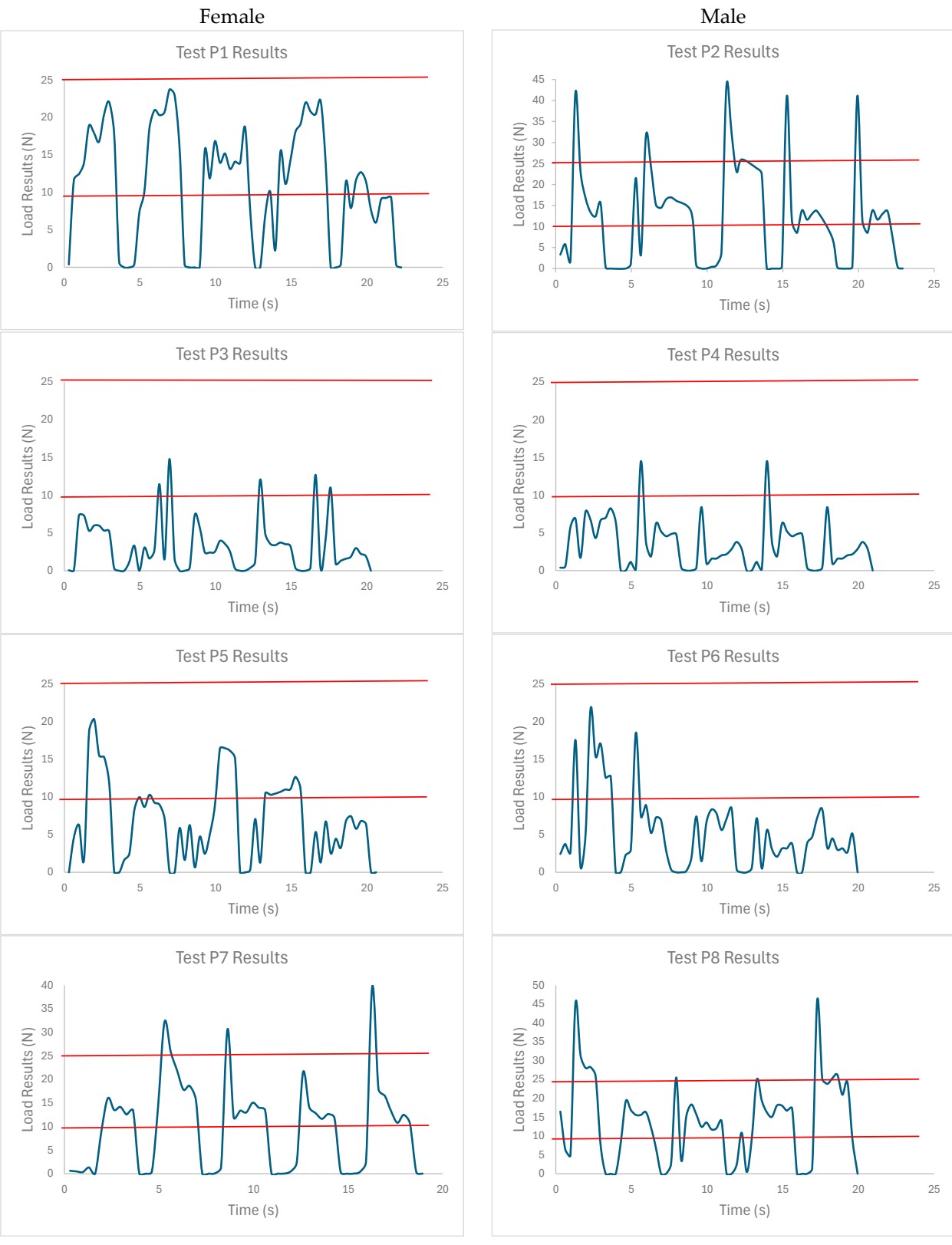

**Figure 5.** Results of eight tests applied to four male and four female users. The force magnitude less than 10 N represents low risk to develop WMSDs, medium risk occurs between 10 N and 25 N and the maximal permissible force is 25 N; therefore, a force magnitude greater than 25 N represents high risk for developing WMSDs.

**Table 4.** Biomechanical assessment of the FR developed by the thumb when the cigarette lighter was activated. The following values were considered: b = 0.016 m, h = 0.01 m, L = 0.029 m, H = 0.277 m and HLV = 0.2451 m.

| Test | $\varepsilon_H$ (μɛ) | $\varepsilon_V$ (μɛ) | $\varepsilon_L$ (μɛ) | $K_{AV}$ (Pa) | Fx (N) | Fy (N) | Fz (N) | AF (N) |
|------|------|------|------|------|------|------|------|------|
| 1F | 79 | 9 | 124 | 70,958,849,724 | 0.35 | −22.45 | 7.66 | 23.73 |
| 2M | 110 | −17 | 174 | 70,958,849,724 | 0.66 | −42.15 | 10.74 | 43.51 |
| 3F | 25 | −12 | 26 | 70,958,849,724 | 0.46 | −12.58 | −1.60 | 12.69 |
| 4M | 34 | −9 | 26 | 70,958,849,724 | 0.34 | −14.41 | −1.60 | 14.50 |
| 5F | 55 | −5 | 77 | 70,958,849,724 | 0.19 | −19.79 | −4.75 | 20.35 |
| 6M | 68 | 6 | 136 | 70,958,849,724 | −0.23 | −19.97 | −8.40 | 21.67 |
| 7F | 98 | −17 | 175 | 70,958,849,724 | 0.66 | −38.24 | −10.81 | 39.74 |
| 8M | 137 | −1 | 151 | 70,958,849,724 | 0.40 | −45.06 | −9.33 | 46.02 |

## 4. Discussion

The key features of the introduced load cell in this study are (a) the possibility of measuring a resulting triaxial force and its components with respect to each reference axis and (b) its configuration, which makes it versatile to be used in different tasks.

To provide the output forces, an analytical model was developed based on linear elasticity theory and solid mechanics models. The model is a combined tension, axial load, bending and torsion problem. The position of the points instrumented with strain gauges has an essential role, first, because the value of the output signal (strains) is required to be relevant for each component of the triaxial force, and second, because the instrumented points, on two sides of the element and in a central position, avoid unwanted signals that could significantly alter the ideal value of the output signal, caused by: the torsion phenomenon, temperature changes or deviations generated by irregularities in the geometry of the mechanical element, or due to effects caused by the strain gauge installation process.

The load cell prototype can be integrated into the structure of the machine where the force is to be measured without affecting the worker's movements through a welding, gluing or screwing process, such as the case presented in this work. Other important characteristics are that, unlike those on the market (cylinder, "S"-type, membrane, etc.), the cell body is easy to manufacture ("L" shape and rectangular section); the instrumentation is clearly defined in this work and its calibration is relatively simple, which may be relevant in engineering applications. Thus, its application can be crucial in situations where commercial triaxial cells, due to their geometric configuration and operating principles, are not feasible to install and apply, making the proposed cell a possible option. However, for some applications, the necessary geometric dimensions and their implementation to the work system may have possible limitations.

Once the calibration factor of the cell was obtained and to evaluate its efficiency, various tests were carried out applying a variety of weights with different orientations, Table 3 and Appendix B.

A demonstrative case of the functionality of the load cell to measure the triaxial forces that are produced when the thumb triggers the cigarette lighter in three directions during the movement of a hand accurately defines the high-risk level associated with the task and the likelihood of developing WMSDs. Eight people (four men and four women) activated a cigarette lighter adapted to the triaxial load cell to measure the AF as a case study. In this first stage, the tests were carried out to seek repeatability and reproducibility of the tests. That is, the same test was repeated with eight different people, activating the lighter under the same laboratory conditions. This made it possible to monitor the triaxial load cell sensitivity and precision with respect to using it as a biomechanical tool to define EH. For example, test P3 female and P4 male had standard deviations of 3.22 and 3.06, respectively, and P1 female, P5 female and P6 male had standard deviations of 5.16, 4.65 and 4.92. These results show that the AF behaviour is too similar between them, despite

the different users' anthropometric characteristics, supporting the idea that the cell can be used as an ergonomic evaluation tool. However, more tests will be required in the future.

As is observed, the 10 N limit was exceeded, clearly indicating that there is a medium-risk to high-risk level of generating WMSDs. However, some limitations were found; for example, the data collection technique must be improved. Therefore, in future work, we will seek to find a technology that allows data to be collected wirelessly, which will allow its direct implementation in the tools or products manipulated in the future. Therefore, the triaxial load cell can have many more applications; for example, it can be integrated to measure the produced forces by dynamic tasks, such as pressing a button during the manual control of some processes, in a second stage of this investigation.

## 5. Conclusions

In this study, a novel load cell prototype was designed to accurately measure the exerted triaxial forces by the right thumb when lighting a lighter. The primary objective was to evaluate the level of ergonomic risk associated with this specific task. By using a mechanical analytical model based on the theory of linear elasticity, the load cell was experimentally calibrated. The average calibration value was found to be $KAV = 70.96 \times 109$ Pa, which when applied in validation tests gave results with differences of less than 3% with respect to the standardized weight applied. Sufficient infrastructure was included to reproduce the prototype, being able to adjust its geometric dimensions and the prototype material to the required capacity and the spaces available in the system where its integration is required.

Due to its configuration and mechanical principles, the adaptability of the load cell as a hand tool during task simulations was evident, showing its versatility in practical applications. In this work, it was applied to measure the exerted forces by the right thumb when activating a lighter. The objective of the test was to evaluate the level of the task ergonomic risk. The experimental results demonstrated that the exerted force could be related to Quervain syndrome.

The prototype will be useful for the development of future research, which will help in the development of better industrial labour practices, in accordance with the capabilities of workers, that help prevent occupational risks and diseases during Industry 4.0 implementations.

**Author Contributions:** Conceptualization, M.A.-F. and M.R.C.-V.; methodology, M.A.-F.; data curation, J.G.V.-A., F.C.-J. and M.L.E.-D.; writing—original draft preparation, M.A.-F., M.R.C.-V. and M.L.E.-D.; supervision, M.R.C.-V. and M.L.E.-D.; Writing—review and editing, M.A.-F., M.R.C.-V. and M.L.E.-D. All authors have read and agreed to the published version of the manuscript.

**Funding:** This research received no external funding.

**Institutional Review Board Statement:** Not applicable.

**Informed Consent Statement:** Not applicable; there is no human experimentation.

**Data Availability Statement:** Appendices B and C include all the data and results obtained in this research.

**Acknowledgments:** The authors of this work thank Universidad Autónoma del Estado de Morelos for supporting the development of this article, and to the students of the methods engineering subject in January 2024.

**Conflicts of Interest:** The authors declare no conflicts of interest, and the investigation does not have funders.

## Appendix A

Calibration Tables.

Tables A1–A3 show the calibration factor determined for each cell; the average characterising each cell is also provided.

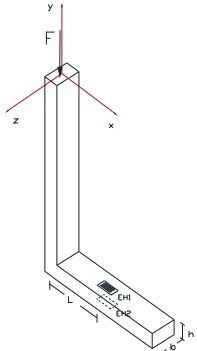

**Figure A1.** Force in the *y*-axis.

**Table A1.** Calibration factor for horizontal cell, using force in the *y*-axis.

| W (N) | $\varepsilon_H$ (με) | b (m) | h (m) | L (m) | I | KH (Pa) |
|---|---|---|---|---|---|---|
| 4.905 | −17 | 0.016 | 0.01 | 0.0327 | $1.3333 \times 10^{-9}$ | 70,761,838,235 |
| 9.81 | −34 | 0.016 | 0.01 | 0.0327 | $1.3333 \times 10^{-9}$ | 70,761,838,235 |
| 14.715 | −51 | 0.016 | 0.01 | 0.0327 | $1.3333 \times 10^{-9}$ | 70,761,838,235 |

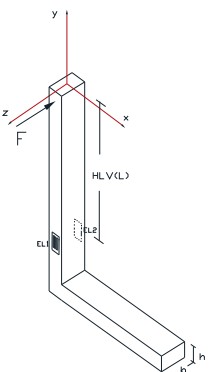

**Figure A2.** Force in the *z*-axis.

**Table A2.** Calibration factor for lateral horizontal cell, using force in the *z*-axis.

| W (N) | $\varepsilon_L$ (με) | b (m) | h (m) | HLV(L) (m) | I | KL |
|---|---|---|---|---|---|---|
| 4.905 | −30 | 0.016 | 0.01 | 0.0925 | $3.4133 \times 10^{-9}$ | 70,892,578,125 |
| 9.81 | −60 | 0.016 | 0.01 | 0.0925 | $3.4133 \times 10^{-9}$ | 70,892,578,125 |

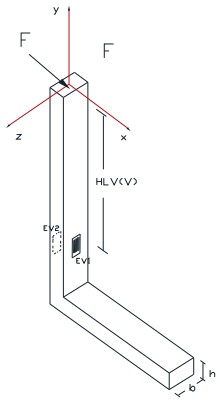

**Figure A3.** Force in the *x*-axis.

**Table A3.** Calibration factor for lateral horizontal cell, using force in the *x*-axis.

| W (N) | $\varepsilon_V$ (με) | b (m) | h (m) | HLV(m) | I | KV |
|-------|------|-------|-------|--------|---|----|
| 4.905 | 48 | 0.016 | 0.01 | 0.0929 | $1.3333 \times 10^{-9}$ | 71,222,132,813 |
| 9.81 | 96 | 0.016 | 0.01 | 0.0929 | $1.3333 \times 10^{-9}$ | 71,222,132,813 |

## Appendix B

Experimental evaluation for an applied force with different angles of α and β.

**Table A4.** Experimental evaluation for an applied force with angles of α = 20° and β = 43°.

| Test 2 | β = 43° | α = 20° | | | | | | | | |
|--------|---------|---------|---|---|---|---|---|---|---|---|
| W (N) | $\varepsilon_H$ (με) | $\varepsilon_V$ (με) | $\varepsilon_L$ (με) | K (Pa) | Fx (N) | Fy (N) | Fz (N) | FR | % of Deviation |
| 0.981 | 3.5 | 4 | 4 | 70,958,849,724 | −0.395 | 0.694 | −0.632 | 1.019 | 3.70 |
| 1.962 | 6 | 7 | 9 | 70,958,849,724 | −0.692 | 1.255 | −1.423 | 2.020 | 2.85 |
| 2.4525 | 7.5 | 9 | 11 | 70,958,849,724 | −0.889 | 1.684 | −1.739 | 2.579 | 4.90 |
| 3.4335 | 10 | 12 | 15 | 70,958,849,724 | −1.186 | 2.245 | −2.371 | 3.474 | 1.17 |
| 4.4145 | 13 | 15 | 20 | 70,958,849,724 | −1.482 | 2.643 | −3.162 | 4.380 | 0.80 |
| 4.905 | 15 | 17 | 22 | 70,958,849,724 | −1.680 | 2.909 | −3.478 | 4.835 | 1.44 |
| 5.886 | 18 | 21 | 27 | 70,958,849,724 | −2.075 | 3.766 | −4.269 | 6.059 | 2.85 |
| 8.3385 | 25 | 29 | 38 | 70,958,849,724 | −2.866 | 5.154 | −6.008 | 8.418 | 0.95 |
| 9.81 | 29.5 | 35 | 44 | 70,958,849,724 | −3.458 | 6.439 | −6.956 | 10.090 | 2.78 |
| 10.791 | 33 | 38 | 48 | 70,958,849,724 | −3.755 | 6.674 | −7.589 | 10.781 | 0.09 |
| 12.2625 | 37.5 | 43 | 55 | 70,958,849,724 | −4.249 | 7.501 | −8.695 | 12.245 | 0.15 |
| 14.715 | 45 | 52 | 66 | 70,958,849,724 | −5.138 | 9.185 | −10.434 | 14.820 | 0.71 |
| 17.1675 | 52.5 | 60 | 77 | 70,958,849,724 | −5.929 | 10.410 | −12.174 | 17.079 | 0.52 |
| 19.62 | 60 | 69 | 88 | 70,958,849,724 | −6.818 | 12.093 | −13.913 | 19.654 | 0.18 |
| 22.0725 | 68 | 77 | 99 | 70,958,849,724 | −7.608 | 13.155 | −15.652 | 21.816 | 1.18 |

**Table A5.** Experimental evaluation for an applied force with angles of α = 25° and β = 43°.

| Test 3 | β = 43° | α = 25° | | | | | | | | |
|--------|---------|---------|---|---|---|---|---|---|---|---|
| W (N) | $\varepsilon_H$ (με) | $\varepsilon_V$ (με) | $\varepsilon_L$ (με) | K (Pa) | Fx (N) | Fy (N) | Fz (N) | FR | % of Deviation |
| 0.981 | 4 | 4 | 4 | 70,958,849,724 | −0.395 | 0.531 | −0.632 | 0.915 | 7.17 |
| 1.962 | 8 | 8 | 8.5 | 70,958,849,724 | −0.790 | 1.062 | −1.344 | 1.886 | 4.02 |
| 2.4525 | 10.5 | 10.5 | 11 | 70,958,849,724 | −1.038 | 1.393 | −1.739 | 2.458 | 0.23 |
| 3.4335 | 15 | 15 | 15 | 70,958,849,724 | −1.482 | 1.991 | −2.371 | 3.433 | 0.02 |
| 4.4145 | 19 | 19 | 19 | 70,958,849,724 | −1.877 | 2.522 | −3.004 | 4.348 | 1.53 |
| 4.905 | 21 | 21 | 21 | 70,958,849,724 | −2.075 | 2.787 | −3.320 | 4.806 | 2.06 |
| 5.886 | 25 | 25 | 26 | 70,958,849,724 | −2.470 | 3.318 | −4.111 | 5.832 | 0.93 |
| 8.3385 | 36 | 36 | 36.5 | 70,958,849,724 | −3.557 | 4.778 | −5.771 | 8.293 | 0.54 |
| 9.81 | 42 | 42 | 43 | 70,958,849,724 | −4.150 | 5.574 | −6.798 | 9.722 | 0.91 |
| 10.791 | 46 | 46 | 48 | 70,958,849,724 | −4.545 | 6.105 | −7.589 | 10.748 | 0.40 |
| 12.2625 | 53 | 53 | 54 | 70,958,849,724 | −5.237 | 7.034 | −8.537 | 12.239 | 0.19 |
| 14.715 | 63 | 63 | 64 | 70,958,849,724 | −6.225 | 8.361 | −10.118 | 14.527 | 1.29 |
| 17.1675 | 74 | 74 | 75 | 70,958,849,724 | −7.312 | 9.821 | −11.857 | 17.044 | 0.72 |
| 19.62 | 84.5 | 84.5 | 86 | 70,958,849,724 | −8.350 | 11.214 | −13.596 | 19.502 | 0.60 |
| 22.0725 | 95 | 95 | 97 | 70,958,849,724 | −9.387 | 12.608 | −15.336 | 21.960 | 0.51 |

**Table A6.** Experimental evaluation for an applied force with angles of α = 0° and β = 50°.

| Test 4 | β = 50° | α = 0° | | | | | | | |
|---|---|---|---|---|---|---|---|---|---|
| W (N) | $\varepsilon_H$ (με) | $\varepsilon_V$ (με) | $\varepsilon_L$ | K (Pa) | Fx (N) | Fy (N) | Fz (N) | FR | % of Deviation |
| 0.981 | −2 | 0 | 4.5 | 70,958,849,724 | 0.000 | 0.652 | −0.711 | 0.965 | 1.62 |
| 1.962 | −5 | −1 | 10 | 70,958,849,724 | 0.099 | 1.172 | −1.581 | 1.971 | 0.44 |
| 2.4525 | −6 | −1 | 12 | 70,958,849,724 | 0.099 | 1.499 | −1.897 | 2.420 | 1.36 |
| 3.4335 | −8 | −1 | 17 | 70,958,849,724 | 0.099 | 2.151 | −2.688 | 3.444 | −0.30 |
| 4.4145 | −10 | −2 | 23 | 70,958,849,724 | 0.198 | 2.345 | −3.636 | 4.331 | 1.93 |
| 4.905 | −10.5 | −2 | 26 | 70,958,849,724 | 0.198 | 2.508 | −4.111 | 4.819 | 1.78 |
| 5.886 | −13 | −2 | 29.5 | 70,958,849,724 | 0.198 | 3.323 | −4.664 | 5.730 | 2.72 |
| 8.3385 | −18.5 | −3 | 42 | 70,958,849,724 | 0.296 | 4.659 | −6.640 | 8.117 | 2.73 |
| 9.81 | −20.5 | −3 | 50.5 | 70,958,849,724 | 0.296 | 5.311 | −7.984 | 9.594 | 2.25 |
| 10.791 | −24.5 | −4 | 54 | 70,958,849,724 | 0.395 | 6.157 | −8.537 | 10.533 | 2.45 |
| 12.2625 | −27 | −4.5 | 62 | 70,958,849,724 | 0.445 | 6.743 | −9.802 | 11.906 | 2.99 |
| 14.715 | −33 | −6 | 75 | 70,958,849,724 | 0.593 | 8.012 | −11.857 | 14.323 | 2.74 |
| 17.1675 | −39 | −7 | 87 | 70,958,849,724 | 0.692 | 9.511 | −13.755 | 16.737 | 2.57 |
| 19.62 | −44 | −8 | 100 | 70,958,849,724 | 0.790 | 10.683 | −15.810 | 19.097 | 2.74 |
| 22.0725 | −50 | −9 | 113 | 70,958,849,724 | 0.889 | 12.182 | −17.865 | 21.641 | 1.99 |

**Appendix C  Coded for Maple 18**

Triaxial Force Cell Model

$$0 = -EH1 + ((1/E)*(-(Fx/(b*h)) - ((6*Fx*H)/(b*h*h)) - ((6*Fy*L)/(b*h*h)))),$$

$$0 = -EV1 + ((1/E)*((Fy/(b*h)) - ((6*Fx*HLV)/(b*h*h)))),$$

$$0 = -EL1 + ((1/E)*((Fy/(b*h)) - ((6*Fz*HLV)/(h*b*b)))),$$

$$0 = -EH2 + ((1/E)*(-(Fx/(b*h)) + ((6*Fx*H)/(b*h*h)) + ((6*Fy*L)/(b*h*h)))),$$

$$0 = -EV2 + ((1/E)*((Fy/(b*h)) + ((6*Fx*HLV)/(b*h*h)))),$$

$$0 = -EL2 + ((1/E)*((Fy/(b*h)) + ((6*Fz*HLV)/(h*b*b)))),$$

$$0 = -(FR*FR) + ((Fx*Fx) + (Fy*Fy) + (Fz*Fz)),$$

$$0 = -EH + (EH1 - EH2),$$

$$0 = -EV + (EV1 - EV2),$$

$$0 = -EL + (EL1 - EL2)$$

Solution Model

$$\{EH1 = (1/12)*(6*EH*HLV + EV*h)/HLV, EH2 = -(1/12)*(6*EH*HLV - EV*h)/HLV, EL1 = -(1/12)*(EH*HLV*h - 6*EL*HLV*L - EV*H*h)/(HLV*L),$$

$$EL2 = -(1/12)*(EH*HLV*h + 6*EL*HLV*L - EV*H*h)/(HLV*L),$$

$$EV1 = -(1/12)*(EH*HLV*h - EV*H*h - 6*EV*HLV*L)/(HLV*L),$$

$$EV2 = -(1/12)*(EH*HLV*h - EV*H*h + 6*EV*HLV*L)/(HLV*L),$$

$$FR = (1/12)*RootOf(-EH^2*HLV^2*h^2 + 2*EH*EV*H*HLV*h^2 - EL^2*L^2*b^2 - EV^2*H^2*h^2 - EV^2*L^2*h^2 + \_Z^2)*h*b*E/(HLV*L),$$

$$Fx = -(1/12)*E*b*h^2*EV/HLV,$$

$$Fy = -(1/12)*E*b*h^2*(EH*HLV - EV*H)/(HLV*L),$$

$$Fz = -(1/12)*E*h*b^2*EL/HLV\}$$

# Appendix D

## Test P1 Female User

**Activation 1**

| $\varepsilon_H$ (µε) | $\varepsilon_V$ (µε) | $\varepsilon_L$ (µε) | K (Pa) | b (m) | h (m) | L (m) | H (m) | HLV (m) | Fx (N) | Fy (N) | Fz (N) | AF (N) |
|---|---|---|---|---|---|---|---|---|---|---|---|---|
| 0 | -1 | 2 | 70,958,849,724 | 0.02 | 0.01 | 0.03 | 0.28 | 0.25 | 0.04 | -0.37 | -0.12 | 0.39 |
| 28 | -2 | 100 | 70,958,849,724 | 0.02 | 0.01 | 0.03 | 0.28 | 0.25 | 0.08 | -9.87 | -6.18 | 11.65 |
| 31 | -3 | 87 | 70,958,849,724 | 0.02 | 0.01 | 0.03 | 0.28 | 0.25 | 0.12 | -11.22 | -5.37 | 12.44 |
| 42 | 4 | 108 | 70,958,849,724 | 0.02 | 0.01 | 0.03 | 0.28 | 0.25 | -0.15 | -12.23 | -6.67 | 13.93 |
| 59 | 5 | 119 | 70,958,849,724 | 0.02 | 0.01 | 0.03 | 0.28 | 0.25 | -0.19 | -17.41 | -7.35 | 18.89 |
| 55 | 4 | 115 | 70,958,849,724 | 0.02 | 0.01 | 0.03 | 0.28 | 0.25 | -0.15 | -16.47 | -7.1 | 17.94 |
| 53 | 5 | 106 | 70,958,849,724 | 0.02 | 0.01 | 0.03 | 0.28 | 0.25 | -0.19 | -15.45 | -6.55 | 16.78 |
| 65 | 6 | 120 | 70,958,849,724 | 0.02 | 0.01 | 0.03 | 0.28 | 0.25 | -0.23 | -18.99 | -7.41 | 20.39 |
| 69 | 5 | 126 | 70,958,849,724 | 0.02 | 0.01 | 0.03 | 0.28 | 0.25 | -0.19 | -20.67 | -7.78 | 22.08 |
| 60 | 8 | 105 | 70,958,849,724 | 0.02 | 0.01 | 0.03 | 0.28 | 0.25 | -0.31 | -16.63 | -6.49 | 17.85 |
| 1 | -1 | 0 | 70,958,849,724 | 0.02 | 0.01 | 0.03 | 0.28 | 0.25 | 0.04 | -0.69 | 0 | 0.7 |
| 0 | 0 | 0 | 70,958,849,724 | 0.02 | 0.01 | 0.03 | 0.28 | 0.25 | 0 | 0 | 0 | 0 |
| 0 | 0 | 0 | 70,958,849,724 | 0.02 | 0.01 | 0.03 | 0.28 | 0.25 | 0 | 0 | 0 | 0 |

**Activation 2**

| $\varepsilon_H$ (µε) | $\varepsilon_V$ (µε) | $\varepsilon_L$ (µε) | K (Pa) | b (m) | h (m) | L (m) | H (m) | HLV (m) | Fx (N) | Fy (N) | Fz (N) | AF (N) |
|---|---|---|---|---|---|---|---|---|---|---|---|---|
| -3 | -2 | 5 | 70,958,849,724 | 0.02 | 0.01 | 0.03 | 0.28 | 0.25 | 0.08 | 0.24 | -0.31 | 0.4 |
| 21 | 6 | 91 | 70,958,849,724 | 0.02 | 0.01 | 0.03 | 0.28 | 0.25 | -0.23 | -4.64 | -5.62 | 7.29 |
| 29 | 4 | 100 | 70,958,849,724 | 0.02 | 0.01 | 0.03 | 0.28 | 0.25 | -0.15 | -7.99 | -6.18 | 10.1 |
| 60 | 7 | 119 | 70,958,849,724 | 0.02 | 0.01 | 0.03 | 0.28 | 0.25 | -0.27 | -16.99 | -7.35 | 18.52 |
| 72 | 11 | 127 | 70,958,849,724 | 0.02 | 0.01 | 0.03 | 0.28 | 0.25 | -0.42 | -19.43 | -7.84 | 20.96 |
| 68 | 8 | 105 | 70,958,849,724 | 0.02 | 0.01 | 0.03 | 0.28 | 0.25 | -0.31 | -19.24 | -6.49 | 20.3 |
| 70 | 10 | 127 | 70,958,849,724 | 0.02 | 0.01 | 0.03 | 0.28 | 0.25 | -0.39 | -19.15 | -7.84 | 20.7 |
| 79 | 9 | 124 | 70,958,849,724 | 0.02 | 0.01 | 0.03 | 0.28 | 0.25 | -0.35 | -22.46 | -7.66 | 23.73 |
| 76 | 8 | 116 | 70,958,849,724 | 0.02 | 0.01 | 0.03 | 0.28 | 0.25 | -0.31 | -21.85 | -7.16 | 22.99 |
| 58 | 11 | 90 | 70,958,849,724 | 0.02 | 0.01 | 0.03 | 0.28 | 0.25 | -0.42 | -14.87 | -5.56 | 15.88 |
| 0 | -1 | -1 | 70,958,849,724 | 0.02 | 0.01 | 0.03 | 0.28 | 0.25 | 0.04 | -0.37 | 0.06 | 0.38 |
| 0 | 0 | 0 | 70,958,849,724 | 0.02 | 0.01 | 0.03 | 0.28 | 0.25 | 0 | 0 | 0 | 0 |
| 0 | 0 | 0 | 70,958,849,724 | 0.02 | 0.01 | 0.03 | 0.28 | 0.25 | 0 | 0 | 0 | 0 |

**Activation 3**

| $\varepsilon_H$ (µε) | $\varepsilon_V$ (µε) | $\varepsilon_L$ (µε) | K (Pa) | b (m) | h (m) | L (m) | H (m) | HLV (m) | Fx (N) | Fy (N) | Fz (N) | AF (N) |
|---|---|---|---|---|---|---|---|---|---|---|---|---|
| -1 | -1 | 0 | 70,958,849,724 | 0.02 | 0.01 | 0.03 | 0.28 | 0.25 | 0.04 | -0.04 | 0 | 0.06 |
| 8 | 6 | 252 | 70,958,849,724 | 0.02 | 0.01 | 0.03 | 0.28 | 0.25 | -0.23 | -0.4 | -15.56 | 15.57 |
| 37 | 9 | 130 | 70,958,849,724 | 0.02 | 0.01 | 0.03 | 0.28 | 0.25 | -0.35 | -8.75 | -8.03 | 11.88 |
| 51 | 3 | 105 | 70,958,849,724 | 0.02 | 0.01 | 0.03 | 0.28 | 0.25 | -0.12 | -15.53 | -6.49 | 16.83 |
| 49 | 9 | 95 | 70,958,849,724 | 0.02 | 0.01 | 0.03 | 0.28 | 0.25 | -0.35 | -12.67 | -5.87 | 13.96 |
| 52 | 8 | 95 | 70,958,849,724 | 0.02 | 0.01 | 0.03 | 0.28 | 0.25 | -0.31 | -14.02 | -5.87 | 15.2 |
| 46 | 9 | 97 | 70,958,849,724 | 0.02 | 0.01 | 0.03 | 0.28 | 0.25 | -0.35 | -11.69 | -5.99 | 13.14 |
| 49 | 9 | 101 | 70,958,849,724 | 0.02 | 0.01 | 0.03 | 0.28 | 0.25 | -0.35 | -12.67 | -6.24 | 14.12 |
| 51 | 12 | 108 | 70,958,849,724 | 0.02 | 0.01 | 0.03 | 0.28 | 0.25 | -0.46 | -12.21 | -6.67 | 13.92 |
| 60 | 6 | 111 | 70,958,849,724 | 0.02 | 0.01 | 0.03 | 0.28 | 0.25 | -0.23 | -17.36 | -6.86 | 18.67 |
| 20 | -3 | 0 | 70,958,849,724 | 0.02 | 0.01 | 0.03 | 0.28 | 0.25 | 0.12 | -7.63 | 0 | 7.63 |
| 0 | 0 | 0 | 70,958,849,724 | 0.02 | 0.01 | 0.03 | 0.28 | 0.25 | 0 | 0 | 0 | 0 |
| 0 | 0 | 0 | 70,958,849,724 | 0.02 | 0.01 | 0.03 | 0.28 | 0.25 | 0 | 0 | 0 | 0 |

**Activation 4**

| $\varepsilon_H$ (µε) | $\varepsilon_V$ (µε) | $\varepsilon_L$ (µε) | K (Pa) | b (m) | h (m) | L (m) | H (m) | HLV (m) | Fx (N) | Fy (N) | Fz (N) | AF (N) |
|---|---|---|---|---|---|---|---|---|---|---|---|---|
| -12 | 1 | 90 | 70,958,849,724 | 0.02 | 0.01 | 0.03 | 0.28 | 0.25 | -0.04 | 4.28 | -5.56 | 7.02 |
| 32 | 1 | 3 | 70,958,849,724 | 0.02 | 0.01 | 0.03 | 0.28 | 0.25 | -0.04 | -10.07 | -0.19 | 10.07 |
| -2 | -8 | 0 | 70,958,849,724 | 0.02 | 0.01 | 0.03 | 0.28 | 0.25 | 0.31 | -2.3 | 0 | 2.32 |
| 46 | 2 | 94 | 70,958,849,724 | 0.02 | 0.01 | 0.03 | 0.28 | 0.25 | -0.08 | -14.27 | -5.81 | 15.41 |
| 36 | 6 | 93 | 70,958,849,724 | 0.02 | 0.01 | 0.03 | 0.28 | 0.25 | -0.23 | -9.53 | -5.74 | 11.13 |
| 47 | 7 | 105 | 70,958,849,724 | 0.02 | 0.01 | 0.03 | 0.28 | 0.25 | -0.27 | -12.75 | -6.49 | 14.31 |
| 59 | 7 | 114 | 70,958,849,724 | 0.02 | 0.01 | 0.03 | 0.28 | 0.25 | -0.27 | -16.67 | -7.04 | 18.1 |
| 60 | 5 | 116 | 70,958,849,724 | 0.02 | 0.01 | 0.03 | 0.28 | 0.25 | -0.19 | -17.73 | -7.16 | 19.12 |
| 69 | 5 | 122 | 70,958,849,724 | 0.02 | 0.01 | 0.03 | 0.28 | 0.25 | -0.19 | -20.67 | -7.53 | 22 |
| 67 | 6 | 110 | 70,958,849,724 | 0.02 | 0.01 | 0.03 | 0.28 | 0.25 | -0.23 | -19.65 | -6.79 | 20.79 |
| 68 | 8 | 111 | 70,958,849,724 | 0.02 | 0.01 | 0.03 | 0.28 | 0.25 | -0.31 | -19.24 | -6.86 | 20.42 |
| 74 | 9 | 124 | 70,958,849,724 | 0.02 | 0.01 | 0.03 | 0.28 | 0.25 | -0.35 | -20.82 | -7.66 | 22.19 |
| 38 | -4 | 0 | 70,958,849,724 | 0.02 | 0.01 | 0.03 | 0.28 | 0.25 | 0.15 | -13.87 | 0 | 13.87 |

**Activation 5**

| $\varepsilon_H$ (µε) | $\varepsilon_V$ (µε) | $\varepsilon_L$ (µε) | K (Pa) | b (m) | h (m) | L (m) | H (m) | HLV (m) | Fx (N) | Fy (N) | Fz (N) | AF (N) |
|---|---|---|---|---|---|---|---|---|---|---|---|---|
| 0 | -1 | 3 | 70,958,849,724 | 0.02 | 0.01 | 0.03 | 0.28 | 0.25 | 0.04 | -0.37 | -0.19 | 0.41 |
| 31 | 0 | 85 | 70,958,849,724 | 0.02 | 0.01 | 0.03 | 0.28 | 0.25 | 0 | -10.11 | -5.25 | 11.4 |
| 25 | 5 | 77 | 70,958,849,724 | 0.02 | 0.01 | 0.03 | 0.28 | 0.25 | -0.19 | -6.31 | -4.76 | 7.91 |
| 36 | 4 | 88 | 70,958,849,724 | 0.02 | 0.01 | 0.03 | 0.28 | 0.25 | -0.15 | -10.27 | -5.44 | 11.62 |
| 45 | 9 | 92 | 70,958,849,724 | 0.02 | 0.01 | 0.03 | 0.28 | 0.25 | -0.35 | -11.36 | -5.68 | 12.71 |
| 45 | 12 | 82 | 70,958,849,724 | 0.02 | 0.01 | 0.03 | 0.28 | 0.25 | -0.46 | -10.26 | -5.06 | 11.45 |
| 33 | 13 | 77 | 70,958,849,724 | 0.02 | 0.01 | 0.03 | 0.28 | 0.25 | -0.5 | -5.97 | -4.76 | 7.65 |
| 29 | 14 | 67 | 70,958,849,724 | 0.02 | 0.01 | 0.03 | 0.28 | 0.25 | -0.54 | -4.3 | -4.14 | 5.99 |
| 41 | 16 | 83 | 70,958,849,724 | 0.02 | 0.01 | 0.03 | 0.28 | 0.25 | -0.62 | -7.48 | -5.13 | 9.09 |
| 40 | 14 | 78 | 70,958,849,724 | 0.02 | 0.01 | 0.03 | 0.28 | 0.25 | -0.54 | -7.89 | -4.82 | 9.26 |
| 41 | 11 | 3 | 70,958,849,724 | 0.02 | 0.01 | 0.03 | 0.28 | 0.25 | -0.42 | -9.32 | -0.19 | 9.33 |
| 0 | -1 | 0 | 70,958,849,724 | 0.02 | 0.01 | 0.03 | 0.28 | 0.25 | 0.04 | -0.37 | 0 | 0.37 |
| 0 | 0 | 0 | 70,958,849,724 | 0.02 | 0.01 | 0.03 | 0.28 | 0.25 | 0 | 0 | 0 | 0 |

## Test P2 Male User

**Activation 1**

| $\varepsilon_H$ (µε) | $\varepsilon_V$ (µε) | $\varepsilon_L$ (µε) | K (Pa) | b (m) | h (m) | L (m) | H (m) | HLV (m) | Fx (N) | Fy (N) | Fz (N) | AF (N) |
|---|---|---|---|---|---|---|---|---|---|---|---|---|
| 0 | -1 | 2 | 70,958,849,724 | 0.02 | 0.01 | 0.03 | 0.28 | 0.25 | 0.04 | -0.37 | -0.12 | 0.39 |
| 28 | -2 | 100 | 70,958,849,724 | 0.02 | 0.01 | 0.03 | 0.28 | 0.25 | 0.08 | -9.87 | -6.18 | 11.65 |
| 31 | -3 | 87 | 70,958,849,724 | 0.02 | 0.01 | 0.03 | 0.28 | 0.25 | 0.12 | -11.22 | -5.37 | 12.44 |
| 42 | 4 | 108 | 70,958,849,724 | 0.02 | 0.01 | 0.03 | 0.28 | 0.25 | -0.15 | -12.23 | -6.67 | 13.93 |
| 59 | 5 | 119 | 70,958,849,724 | 0.02 | 0.01 | 0.03 | 0.28 | 0.25 | -0.19 | -17.41 | -7.35 | 18.89 |
| 55 | 4 | 115 | 70,958,849,724 | 0.02 | 0.01 | 0.03 | 0.28 | 0.25 | -0.15 | -16.47 | -7.1 | 17.94 |
| 53 | 5 | 106 | 70,958,849,724 | 0.02 | 0.01 | 0.03 | 0.28 | 0.25 | -0.19 | -15.45 | -6.55 | 16.78 |
| 65 | 6 | 120 | 70,958,849,724 | 0.02 | 0.01 | 0.03 | 0.28 | 0.25 | -0.23 | -18.99 | -7.41 | 20.39 |
| 69 | 5 | 126 | 70,958,849,724 | 0.02 | 0.01 | 0.03 | 0.28 | 0.25 | -0.19 | -20.67 | -7.78 | 22.08 |
| 60 | 8 | 105 | 70,958,849,724 | 0.02 | 0.01 | 0.03 | 0.28 | 0.25 | -0.31 | -16.63 | -6.49 | 17.85 |
| 1 | -1 | 0 | 70,958,849,724 | 0.02 | 0.01 | 0.03 | 0.28 | 0.25 | 0.04 | -0.69 | 0 | 0.7 |
| 0 | 0 | 0 | 70,958,849,724 | 0.02 | 0.01 | 0.03 | 0.28 | 0.25 | 0 | 0 | 0 | 0 |
| 0 | 0 | 0 | 70,958,849,724 | 0.02 | 0.01 | 0.03 | 0.28 | 0.25 | 0 | 0 | 0 | 0 |

**Activation 2**

| $\varepsilon_H$ (µε) | $\varepsilon_V$ (µε) | $\varepsilon_L$ (µε) | K (Pa) | b (m) | h (m) | L (m) | H (m) | HLV (m) | Fx (N) | Fy (N) | Fz (N) | AF (N) |
|---|---|---|---|---|---|---|---|---|---|---|---|---|
| -3 | -2 | 5 | 70,958,849,724 | 0.02 | 0.01 | 0.03 | 0.28 | 0.25 | 0.08 | 0.24 | -0.31 | 0.4 |
| 21 | 6 | 91 | 70,958,849,724 | 0.02 | 0.01 | 0.03 | 0.28 | 0.25 | -0.23 | -4.64 | -5.62 | 7.29 |
| 29 | 4 | 100 | 70,958,849,724 | 0.02 | 0.01 | 0.03 | 0.28 | 0.25 | -0.15 | -7.99 | -6.18 | 10.1 |
| 60 | 7 | 119 | 70,958,849,724 | 0.02 | 0.01 | 0.03 | 0.28 | 0.25 | -0.27 | -16.99 | -7.35 | 18.52 |
| 72 | 11 | 127 | 70,958,849,724 | 0.02 | 0.01 | 0.03 | 0.28 | 0.25 | -0.42 | -19.43 | -7.84 | 20.96 |
| 68 | 8 | 105 | 70,958,849,724 | 0.02 | 0.01 | 0.03 | 0.28 | 0.25 | -0.31 | -19.24 | -6.49 | 20.3 |
| 70 | 10 | 127 | 70,958,849,724 | 0.02 | 0.01 | 0.03 | 0.28 | 0.25 | -0.39 | -19.15 | -7.84 | 20.7 |
| 79 | 9 | 124 | 70,958,849,724 | 0.02 | 0.01 | 0.03 | 0.28 | 0.25 | -0.35 | -22.46 | -7.66 | 23.73 |
| 76 | 8 | 116 | 70,958,849,724 | 0.02 | 0.01 | 0.03 | 0.28 | 0.25 | -0.31 | -21.85 | -7.16 | 22.99 |
| 58 | 11 | 90 | 70,958,849,724 | 0.02 | 0.01 | 0.03 | 0.28 | 0.25 | -0.42 | -14.87 | -5.56 | 15.88 |
| 0 | -1 | -1 | 70,958,849,724 | 0.02 | 0.01 | 0.03 | 0.28 | 0.25 | 0.04 | -0.37 | 0.06 | 0.38 |
| 0 | 0 | 0 | 70,958,849,724 | 0.02 | 0.01 | 0.03 | 0.28 | 0.25 | 0 | 0 | 0 | 0 |
| 0 | 0 | 0 | 70,958,849,724 | 0.02 | 0.01 | 0.03 | 0.28 | 0.25 | 0 | 0 | 0 | 0 |

**Activation 3**

| $\varepsilon_H$ (µε) | $\varepsilon_V$ (µε) | $\varepsilon_L$ (µε) | K (Pa) | b (m) | h (m) | L (m) | H (m) | HLV (m) | Fx (N) | Fy (N) | Fz (N) | AF (N) |
|---|---|---|---|---|---|---|---|---|---|---|---|---|
| -1 | -1 | 0 | 70,958,849,724 | 0.02 | 0.01 | 0.03 | 0.28 | 0.25 | 0.04 | -0.04 | 0 | 0.06 |
| 8 | 6 | 252 | 70,958,849,724 | 0.02 | 0.01 | 0.03 | 0.28 | 0.25 | -0.23 | -0.4 | -15.56 | 15.57 |
| 37 | 9 | 130 | 70,958,849,724 | 0.02 | 0.01 | 0.03 | 0.28 | 0.25 | -0.35 | -8.75 | -8.03 | 11.88 |
| 51 | 3 | 105 | 70,958,849,724 | 0.02 | 0.01 | 0.03 | 0.28 | 0.25 | -0.12 | -15.53 | -6.49 | 16.83 |
| 49 | 9 | 95 | 70,958,849,724 | 0.02 | 0.01 | 0.03 | 0.28 | 0.25 | -0.35 | -12.67 | -5.87 | 13.96 |
| 52 | 8 | 95 | 70,958,849,724 | 0.02 | 0.01 | 0.03 | 0.28 | 0.25 | -0.31 | -14.02 | -5.87 | 15.2 |
| 46 | 9 | 97 | 70,958,849,724 | 0.02 | 0.01 | 0.03 | 0.28 | 0.25 | -0.35 | -11.69 | -5.99 | 13.14 |
| 49 | 9 | 101 | 70,958,849,724 | 0.02 | 0.01 | 0.03 | 0.28 | 0.25 | -0.35 | -12.67 | -6.24 | 14.12 |
| 51 | 12 | 108 | 70,958,849,724 | 0.02 | 0.01 | 0.03 | 0.28 | 0.25 | -0.46 | -12.21 | -6.67 | 13.92 |
| 60 | 6 | 111 | 70,958,849,724 | 0.02 | 0.01 | 0.03 | 0.28 | 0.25 | -0.23 | -17.36 | -6.86 | 18.67 |
| 20 | -3 | 0 | 70,958,849,724 | 0.02 | 0.01 | 0.03 | 0.28 | 0.25 | 0.12 | -7.63 | 0 | 7.63 |
| 0 | 0 | 0 | 70,958,849,724 | 0.02 | 0.01 | 0.03 | 0.28 | 0.25 | 0 | 0 | 0 | 0 |
| 0 | 0 | 0 | 70,958,849,724 | 0.02 | 0.01 | 0.03 | 0.28 | 0.25 | 0 | 0 | 0 | 0 |

**Activation 4**

| $\varepsilon_H$ (µε) | $\varepsilon_V$ (µε) | $\varepsilon_L$ (µε) | K (Pa) | b (m) | h (m) | L (m) | H (m) | HLV (m) | Fx (N) | Fy (N) | Fz (N) | AF (N) |
|---|---|---|---|---|---|---|---|---|---|---|---|---|
| -12 | 1 | 90 | 70,958,849,724 | 0.02 | 0.01 | 0.03 | 0.28 | 0.25 | -0.04 | 4.28 | -5.56 | 7.02 |
| 32 | 1 | 3 | 70,958,849,724 | 0.02 | 0.01 | 0.03 | 0.28 | 0.25 | -0.04 | -10.07 | -0.19 | 10.07 |
| -2 | -8 | 0 | 70,958,849,724 | 0.02 | 0.01 | 0.03 | 0.28 | 0.25 | 0.31 | -2.3 | 0 | 2.32 |
| 46 | 2 | 94 | 70,958,849,724 | 0.02 | 0.01 | 0.03 | 0.28 | 0.25 | -0.08 | -14.27 | -5.81 | 15.41 |
| 36 | 6 | 93 | 70,958,849,724 | 0.02 | 0.01 | 0.03 | 0.28 | 0.25 | -0.23 | -9.53 | -5.74 | 11.13 |
| 47 | 7 | 105 | 70,958,849,724 | 0.02 | 0.01 | 0.03 | 0.28 | 0.25 | -0.27 | -12.75 | -6.49 | 14.31 |
| 59 | 7 | 114 | 70,958,849,724 | 0.02 | 0.01 | 0.03 | 0.28 | 0.25 | -0.27 | -16.67 | -7.04 | 18.1 |
| 60 | 5 | 116 | 70,958,849,724 | 0.02 | 0.01 | 0.03 | 0.28 | 0.25 | -0.19 | -17.73 | -7.16 | 19.12 |
| 69 | 5 | 122 | 70,958,849,724 | 0.02 | 0.01 | 0.03 | 0.28 | 0.25 | -0.19 | -20.67 | -7.53 | 22 |
| 67 | 6 | 110 | 70,958,849,724 | 0.02 | 0.01 | 0.03 | 0.28 | 0.25 | -0.23 | -19.65 | -6.79 | 20.79 |
| 68 | 8 | 111 | 70,958,849,724 | 0.02 | 0.01 | 0.03 | 0.28 | 0.25 | -0.31 | -19.24 | -6.86 | 20.42 |
| 74 | 9 | 124 | 70,958,849,724 | 0.02 | 0.01 | 0.03 | 0.28 | 0.25 | -0.35 | -20.82 | -7.66 | 22.19 |
| 38 | -4 | 0 | 70,958,849,724 | 0.02 | 0.01 | 0.03 | 0.28 | 0.25 | 0.15 | -13.87 | 0 | 13.87 |

**Activation 5**

| $\varepsilon_H$ (µε) | $\varepsilon_V$ (µε) | $\varepsilon_L$ (µε) | K (Pa) | b (m) | h (m) | L (m) | H (m) | HLV (m) | Fx (N) | Fy (N) | Fz (N) | AF (N) |
|---|---|---|---|---|---|---|---|---|---|---|---|---|
| 0 | -1 | 3 | 70,958,849,724 | 0.02 | 0.01 | 0.03 | 0.28 | 0.25 | 0.04 | -0.37 | -0.19 | 0.41 |
| 31 | 0 | 85 | 70,958,849,724 | 0.02 | 0.01 | 0.03 | 0.28 | 0.25 | 0 | -10.11 | -5.25 | 11.4 |
| 25 | 5 | 77 | 70,958,849,724 | 0.02 | 0.01 | 0.03 | 0.28 | 0.25 | -0.19 | -6.31 | -4.76 | 7.91 |
| 36 | 4 | 88 | 70,958,849,724 | 0.02 | 0.01 | 0.03 | 0.28 | 0.25 | -0.15 | -10.27 | -5.44 | 11.62 |
| 45 | 9 | 92 | 70,958,849,724 | 0.02 | 0.01 | 0.03 | 0.28 | 0.25 | -0.35 | -11.36 | -5.68 | 12.71 |
| 45 | 12 | 82 | 70,958,849,724 | 0.02 | 0.01 | 0.03 | 0.28 | 0.25 | -0.46 | -10.26 | -5.06 | 11.45 |
| 33 | 13 | 77 | 70,958,849,724 | 0.02 | 0.01 | 0.03 | 0.28 | 0.25 | -0.5 | -5.97 | -4.76 | 7.65 |
| 29 | 14 | 67 | 70,958,849,724 | 0.02 | 0.01 | 0.03 | 0.28 | 0.25 | -0.54 | -4.3 | -4.14 | 5.99 |
| 41 | 16 | 83 | 70,958,849,724 | 0.02 | 0.01 | 0.03 | 0.28 | 0.25 | -0.62 | -7.48 | -5.13 | 9.09 |
| 40 | 14 | 78 | 70,958,849,724 | 0.02 | 0.01 | 0.03 | 0.28 | 0.25 | -0.54 | -7.89 | -4.82 | 9.26 |
| 41 | 11 | 3 | 70,958,849,724 | 0.02 | 0.01 | 0.03 | 0.28 | 0.25 | -0.42 | -9.32 | -0.19 | 9.33 |
| 0 | -1 | 0 | 70,958,849,724 | 0.02 | 0.01 | 0.03 | 0.28 | 0.25 | 0.04 | -0.37 | 0 | 0.37 |
| 0 | 0 | 0 | 70,958,849,724 | 0.02 | 0.01 | 0.03 | 0.28 | 0.25 | 0 | 0 | 0 | 0 |

## Test P3 Female User

**Activation 1**

| $\varepsilon_H$ (με) | $\varepsilon_V$ (με) | $\varepsilon_L$ (με) | K (Pa) | b (m) | h (m) | L (m) | H (m) | HLV (m) | Fx (N) | Fy (N) | Fz (N) | AF (N) |
|---|---|---|---|---|---|---|---|---|---|---|---|---|
| -1 | -1 | 0 | 70,958,849,724 | 0.02 | 0.01 | 0.03 | 0.28 | 0.25 | 0.04 | -0.04 | 0 | 0.06 |
| -1 | -1 | 0 | 70,958,849,724 | 0.02 | 0.01 | 0.03 | 0.28 | 0.25 | 0.04 | -0.04 | 0 | 0.06 |
| 15 | -5 | 46 | 70,958,849,724 | 0.02 | 0.01 | 0.03 | 0.28 | 0.25 | 0.19 | -6.74 | -2.84 | 7.31 |
| 15 | -5 | 46 | 70,958,849,724 | 0.02 | 0.01 | 0.03 | 0.28 | 0.25 | 0.19 | -6.74 | -2.84 | 7.31 |
| 5 | -9 | 30 | 70,958,849,724 | 0.02 | 0.01 | 0.03 | 0.28 | 0.25 | 0.35 | -4.95 | -1.85 | 5.3 |
| 8 | -8 | 34 | 70,958,849,724 | 0.02 | 0.01 | 0.03 | 0.28 | 0.25 | 0.31 | -5.56 | -2.1 | 5.95 |
| 8 | -8 | 34 | 70,958,849,724 | 0.02 | 0.01 | 0.03 | 0.28 | 0.25 | 0.31 | -5.56 | -2.1 | 5.95 |
| 5 | -9 | 30 | 70,958,849,724 | 0.02 | 0.01 | 0.03 | 0.28 | 0.25 | 0.35 | -4.95 | -1.85 | 5.3 |
| 5 | -9 | 27 | 70,958,849,724 | 0.02 | 0.01 | 0.03 | 0.28 | 0.25 | 0.35 | -4.95 | -1.67 | 5.23 |
| -1 | 0 | 0 | 70,958,849,724 | 0.02 | 0.01 | 0.03 | 0.28 | 0.25 | 0 | 0.33 | 0 | 0.33 |
| 0 | 0 | 0 | 70,958,849,724 | 0.02 | 0.01 | 0.03 | 0.28 | 0.25 | 0 | 0 | 0 | 0 |
| 0 | 0 | 0 | 70,958,849,724 | 0.02 | 0.01 | 0.03 | 0.28 | 0.25 | 0 | 0 | 0 | 0 |
| 0 | 0 | 0 | 70,958,849,724 | 0.02 | 0.01 | 0.03 | 0.28 | 0.25 | 0 | 0 | 0 | 0 |

**Activation 2**

| $\varepsilon_H$ (με) | $\varepsilon_V$ (με) | $\varepsilon_L$ (με) | K (Pa) | b (m) | h (m) | L (m) | H (m) | HLV (m) | Fx (N) | Fy (N) | Fz (N) | AF (N) |
|---|---|---|---|---|---|---|---|---|---|---|---|---|
| 0 | 0 | 0 | 70,958,849,724 | 0.02 | 0.01 | 0.03 | 0.28 | 0.25 | 0 | 0 | 0 | 0 |
| -17 | -15 | 16 | 70,958,849,724 | 0.02 | 0.01 | 0.03 | 0.28 | 0.25 | 0.58 | 0.02 | -0.99 | 1.15 |
| 9 | -1 | 0 | 70,958,849,724 | 0.02 | 0.01 | 0.03 | 0.28 | 0.25 | 0.04 | -3.3 | 0 | 3.31 |
| 0 | 0 | 0 | 70,958,849,724 | 0.02 | 0.01 | 0.03 | 0.28 | 0.25 | 0 | 0 | 0 | 0 |
| -10 | -9 | 49 | 70,958,849,724 | 0.02 | 0.01 | 0.03 | 0.28 | 0.25 | 0.35 | -0.06 | -3.03 | 3.05 |
| -6 | -1 | 3 | 70,958,849,724 | 0.02 | 0.01 | 0.03 | 0.28 | 0.25 | 0.04 | 1.59 | -0.19 | 1.6 |
| -9 | -3 | -32 | 70,958,849,724 | 0.02 | 0.01 | 0.03 | 0.28 | 0.25 | 0.12 | 1.83 | 1.98 | 2.7 |
| 43 | 7 | 5 | 70,958,849,724 | 0.02 | 0.01 | 0.03 | 0.28 | 0.25 | -0.27 | -11.45 | -0.31 | 11.46 |
| 0 | -4 | -2 | 70,958,849,724 | 0.02 | 0.01 | 0.03 | 0.28 | 0.25 | 0.15 | -1.47 | 0.12 | 1.49 |
| 28 | -11 | 108 | 70,958,849,724 | 0.02 | 0.01 | 0.03 | 0.28 | 0.25 | 0.42 | -13.19 | -6.67 | 14.79 |
| 6 | 1 | 0 | 70,958,849,724 | 0.02 | 0.01 | 0.03 | 0.28 | 0.25 | -0.04 | -1.59 | 0 | 1.59 |
| 0 | 0 | 0 | 70,958,849,724 | 0.02 | 0.01 | 0.03 | 0.28 | 0.25 | 0 | 0 | 0 | 0 |
| 0 | 0 | 0 | 70,958,849,724 | 0.02 | 0.01 | 0.03 | 0.28 | 0.25 | 0 | 0 | 0 | 0 |

**Activation 3**

| $\varepsilon_H$ (με) | $\varepsilon_V$ (με) | $\varepsilon_L$ (με) | K (Pa) | b (m) | h (m) | L (m) | H (m) | HLV (m) | Fx (N) | Fy (N) | Fz (N) | AF (N) |
|---|---|---|---|---|---|---|---|---|---|---|---|---|
| 0 | -1 | 0 | 70,958,849,724 | 0.02 | 0.01 | 0.03 | 0.28 | 0.25 | 0.04 | -0.37 | 0 | 0.37 |
| -37 | -18 | 79 | 70,958,849,724 | 0.02 | 0.01 | 0.03 | 0.28 | 0.25 | 0.69 | 5.43 | -4.88 | 7.34 |
| 13 | -3 | 33 | 70,958,849,724 | 0.02 | 0.01 | 0.03 | 0.28 | 0.25 | 0.12 | -5.35 | -2.04 | 5.72 |
| -2 | -6 | 30 | 70,958,849,724 | 0.02 | 0.01 | 0.03 | 0.28 | 0.25 | 0.23 | -1.56 | -1.85 | 2.43 |
| -1 | -6 | 24 | 70,958,849,724 | 0.02 | 0.01 | 0.03 | 0.28 | 0.25 | 0.23 | -1.89 | -1.48 | 2.41 |
| -1 | -6 | 26 | 70,958,849,724 | 0.02 | 0.01 | 0.03 | 0.28 | 0.25 | 0.23 | -1.89 | -1.61 | 2.49 |
| 4 | -6 | 28 | 70,958,849,724 | 0.02 | 0.01 | 0.03 | 0.28 | 0.25 | 0.23 | -3.52 | -1.73 | 3.93 |
| 3 | -6 | 25 | 70,958,849,724 | 0.02 | 0.01 | 0.03 | 0.28 | 0.25 | 0.23 | -3.19 | -1.54 | 3.55 |
| 0 | -6 | 20 | 70,958,849,724 | 0.02 | 0.01 | 0.03 | 0.28 | 0.25 | 0.23 | -2.21 | -1.24 | 2.54 |
| -6 | -5 | 3 | 70,958,849,724 | 0.02 | 0.01 | 0.03 | 0.28 | 0.25 | 0.19 | 0.11 | -0.19 | 0.29 |
| 0 | 0 | 0 | 70,958,849,724 | 0.02 | 0.01 | 0.03 | 0.28 | 0.25 | 0 | 0 | 0 | 0 |
| 0 | 0 | 0 | 70,958,849,724 | 0.02 | 0.01 | 0.03 | 0.28 | 0.25 | 0 | 0 | 0 | 0 |
| 0 | 0 | 0 | 70,958,849,724 | 0.02 | 0.01 | 0.03 | 0.28 | 0.25 | 0 | 0 | 0 | 0 |

**Activation 4**

| $\varepsilon_H$ (με) | $\varepsilon_V$ (με) | $\varepsilon_L$ (με) | K (Pa) | b (m) | h (m) | L (m) | H (m) | HLV (m) | Fx (N) | Fy (N) | Fz (N) | AF (N) |
|---|---|---|---|---|---|---|---|---|---|---|---|---|
| 0 | -1 | 0 | 70,958,849,724 | 0.02 | 0.01 | 0.03 | 0.28 | 0.25 | 0.04 | -0.37 | 0 | 0.37 |
| 2 | 1 | 18 | 70,958,849,724 | 0.02 | 0.01 | 0.03 | 0.28 | 0.25 | -0.04 | -0.28 | -1.11 | 1.15 |
| 30 | -3 | 83 | 70,958,849,724 | 0.02 | 0.01 | 0.03 | 0.28 | 0.25 | 0.12 | -10.89 | -5.13 | 12.04 |
| 10 | -2 | 47 | 70,958,849,724 | 0.02 | 0.01 | 0.03 | 0.28 | 0.25 | 0.08 | -4 | -2.9 | 4.94 |
| 5 | -3 | 37 | 70,958,849,724 | 0.02 | 0.01 | 0.03 | 0.28 | 0.25 | 0.12 | -2.74 | -2.29 | 3.57 |
| 5 | -2 | 39 | 70,958,849,724 | 0.02 | 0.01 | 0.03 | 0.28 | 0.25 | 0.08 | -2.37 | -2.41 | 3.38 |
| 7 | -2 | 35 | 70,958,849,724 | 0.02 | 0.01 | 0.03 | 0.28 | 0.25 | 0.08 | -3.02 | -2.16 | 3.72 |
| 7 | -1 | 37 | 70,958,849,724 | 0.02 | 0.01 | 0.03 | 0.28 | 0.25 | 0.04 | -2.65 | -2.29 | 3.5 |
| 6 | -2 | 30 | 70,958,849,724 | 0.02 | 0.01 | 0.03 | 0.28 | 0.25 | 0.08 | -2.69 | -1.85 | 3.27 |
| 0 | -1 | 0 | 70,958,849,724 | 0.02 | 0.01 | 0.03 | 0.28 | 0.25 | 0.04 | -0.37 | 0 | 0.37 |
| 0 | 0 | 0 | 70,958,849,724 | 0.02 | 0.01 | 0.03 | 0.28 | 0.25 | 0 | 0 | 0 | 0 |
| 0 | 0 | 0 | 70,958,849,724 | 0.02 | 0.01 | 0.03 | 0.28 | 0.25 | 0 | 0 | 0 | 0 |
| 0 | 0 | 0 | 70,958,849,724 | 0.02 | 0.01 | 0.03 | 0.28 | 0.25 | 0 | 0 | 0 | 0 |

**Activation 5**

| $\varepsilon_H$ (με) | $\varepsilon_V$ (με) | $\varepsilon_L$ (με) | K (Pa) | b (m) | h (m) | L (m) | H (m) | HLV (m) | Fx (N) | Fy (N) | Fz (N) | AF (N) |
|---|---|---|---|---|---|---|---|---|---|---|---|---|
| 0 | -1 | 3 | 70,958,849,724 | 0.02 | 0.01 | 0.03 | 0.28 | 0.25 | 0.04 | -0.37 | -0.19 | 0.41 |
| 25 | -12 | 26 | 70,958,849,724 | 0.02 | 0.01 | 0.03 | 0.28 | 0.25 | 0.46 | -12.58 | -1.61 | 12.69 |
| -1 | 0 | 0 | 70,958,849,724 | 0.02 | 0.01 | 0.03 | 0.28 | 0.25 | 0 | 0.33 | 0 | 0.33 |
| -23 | -31 | -7 | 70,958,849,724 | 0.02 | 0.01 | 0.03 | 0.28 | 0.25 | 1.2 | -3.93 | 0.43 | 4.13 |
| 17 | -14 | 39 | 70,958,849,724 | 0.02 | 0.01 | 0.03 | 0.28 | 0.25 | 0.54 | -10.71 | -2.41 | 10.99 |
| -12 | -11 | 13 | 70,958,849,724 | 0.02 | 0.01 | 0.03 | 0.28 | 0.25 | 0.42 | -0.14 | -0.8 | 0.92 |
| -8 | -9 | 17 | 70,958,849,724 | 0.02 | 0.01 | 0.03 | 0.28 | 0.25 | 0.35 | -0.71 | -1.05 | 1.31 |
| -6 | -8 | 19 | 70,958,849,724 | 0.02 | 0.01 | 0.03 | 0.28 | 0.25 | 0.31 | -0.99 | -1.17 | 1.57 |
| -5 | -8 | 20 | 70,958,849,724 | 0.02 | 0.01 | 0.03 | 0.28 | 0.25 | 0.31 | -1.32 | -1.24 | 1.83 |
| -2 | -9 | 21 | 70,958,849,724 | 0.02 | 0.01 | 0.03 | 0.28 | 0.25 | 0.35 | -2.67 | -1.3 | 2.98 |
| -5 | -10 | 12 | 70,958,849,724 | 0.02 | 0.01 | 0.03 | 0.28 | 0.25 | 0.39 | -2.06 | -0.74 | 2.22 |
| -7 | -1 | 0 | 70,958,849,724 | 0.02 | 0.01 | 0.03 | 0.28 | 0.25 | 0.04 | 1.92 | 0 | 1.92 |
| 0 | 0 | 0 | 70,958,849,724 | 0.02 | 0.01 | 0.03 | 0.28 | 0.25 | 0 | 0 | 0 | 0 |

## Test P4 Male User

**Activation 1**

| $\varepsilon_H$ (με) | $\varepsilon_V$ (με) | $\varepsilon_L$ (με) | K (Pa) | b (m) | h (m) | L (m) | H (m) | HLV (m) | Fx (N) | Fy (N) | Fz (N) | AF (N) |
|---|---|---|---|---|---|---|---|---|---|---|---|---|
| 0 | -1 | -1 | 70,958,849,724 | 0.02 | 0.01 | 0.03 | 0.28 | 0.25 | 0.04 | -0.37 | 0.06 | 0.38 |
| 0 | 0 | 9 | 70,958,849,724 | 0.02 | 0.01 | 0.03 | 0.28 | 0.25 | 0 | 0 | -0.56 | 0.56 |
| -15 | 2 | 16 | 70,958,849,724 | 0.02 | 0.01 | 0.03 | 0.28 | 0.25 | -0.08 | 5.63 | -0.99 | 5.72 |
| -26 | -5 | 29 | 70,958,849,724 | 0.02 | 0.01 | 0.03 | 0.28 | 0.25 | 0.19 | 6.64 | -1.79 | 6.88 |
| 4 | -1 | 2 | 70,958,849,724 | 0.02 | 0.01 | 0.03 | 0.28 | 0.25 | 0.04 | -1.67 | -0.12 | 1.68 |
| -22 | 1 | 32 | 70,958,849,724 | 0.02 | 0.01 | 0.03 | 0.28 | 0.25 | -0.04 | 7.55 | -1.98 | 7.8 |
| 17 | -2 | 38 | 70,958,849,724 | 0.02 | 0.01 | 0.03 | 0.28 | 0.25 | 0.08 | -6.28 | -2.35 | 6.71 |
| 8 | -3 | 35 | 70,958,849,724 | 0.02 | 0.01 | 0.03 | 0.28 | 0.25 | 0.12 | -3.72 | -2.16 | 4.3 |
| 12 | -6 | 42 | 70,958,849,724 | 0.02 | 0.01 | 0.03 | 0.28 | 0.25 | 0.23 | -6.13 | -2.59 | 6.66 |
| 14 | -5 | 45 | 70,958,849,724 | 0.02 | 0.01 | 0.03 | 0.28 | 0.25 | 0.19 | -6.41 | -2.78 | 6.99 |
| 17 | -6 | 45 | 70,958,849,724 | 0.02 | 0.01 | 0.03 | 0.28 | 0.25 | 0.23 | -7.76 | -2.78 | 8.24 |
| 13 | -5 | 35 | 70,958,849,724 | 0.02 | 0.01 | 0.03 | 0.28 | 0.25 | 0.19 | -6.08 | -2.16 | 6.46 |
| 0 | 0 | 0 | 70,958,849,724 | 0.02 | 0.01 | 0.03 | 0.28 | 0.25 | 0 | 0 | 0 | 0 |

**Activation 2**

| $\varepsilon_H$ (με) | $\varepsilon_V$ (με) | $\varepsilon_L$ (με) | K (Pa) | b (m) | h (m) | L (m) | H (m) | HLV (m) | Fx (N) | Fy (N) | Fz (N) | AF (N) |
|---|---|---|---|---|---|---|---|---|---|---|---|---|
| 0 | -3 | 4 | 70,958,849,724 | 0.02 | 0.01 | 0.03 | 0.28 | 0.25 | 0.12 | -1.11 | -0.25 | 1.14 |
| -1 | -1 | 3 | 70,958,849,724 | 0.02 | 0.01 | 0.03 | 0.28 | 0.25 | 0.04 | -0.04 | -0.19 | 0.19 |
| 34 | -9 | 26 | 70,958,849,724 | 0.02 | 0.01 | 0.03 | 0.28 | 0.25 | 0.35 | -14.41 | -1.61 | 14.5 |
| -2 | -12 | -4 | 70,958,849,724 | 0.02 | 0.01 | 0.03 | 0.28 | 0.25 | 0.46 | -3.77 | 0.25 | 3.81 |
| -7 | -5 | 29 | 70,958,849,724 | 0.02 | 0.01 | 0.03 | 0.28 | 0.25 | 0.19 | 0.44 | -1.79 | 1.85 |
| 9 | -8 | 33 | 70,958,849,724 | 0.02 | 0.01 | 0.03 | 0.28 | 0.25 | 0.31 | -5.89 | -2.04 | 6.24 |
| 8 | -6 | 31 | 70,958,849,724 | 0.02 | 0.01 | 0.03 | 0.28 | 0.25 | 0.23 | -4.82 | -1.91 | 5.19 |
| 7 | -5 | 31 | 70,958,849,724 | 0.02 | 0.01 | 0.03 | 0.28 | 0.25 | 0.19 | -4.13 | -1.91 | 4.55 |
| 8 | -5 | 30 | 70,958,849,724 | 0.02 | 0.01 | 0.03 | 0.28 | 0.25 | 0.19 | -4.45 | -1.85 | 4.83 |
| 8 | -5 | 29 | 70,958,849,724 | 0.02 | 0.01 | 0.03 | 0.28 | 0.25 | 0.19 | -4.45 | -1.79 | 4.8 |
| 0 | -1 | 0 | 70,958,849,724 | 0.02 | 0.01 | 0.03 | 0.28 | 0.25 | 0.04 | -0.37 | 0 | 0.37 |
| 0 | 0 | 0 | 70,958,849,724 | 0.02 | 0.01 | 0.03 | 0.28 | 0.25 | 0 | 0 | 0 | 0 |
| 0 | 0 | 0 | 70,958,849,724 | 0.02 | 0.01 | 0.03 | 0.28 | 0.25 | 0 | 0 | 0 | 0 |

**Activation 3**

| $\varepsilon_H$ (με) | $\varepsilon_V$ (με) | $\varepsilon_L$ (με) | K (Pa) | b (m) | h (m) | L (m) | H (m) | HLV (m) | Fx (N) | Fy (N) | Fz (N) | AF (N) |
|---|---|---|---|---|---|---|---|---|---|---|---|---|
| 1 | 0 | 3 | 70,958,849,724 | 0.02 | 0.01 | 0.03 | 0.28 | 0.25 | 0 | -0.33 | -0.19 | 0.38 |
| -25 | 0 | 34 | 70,958,849,724 | 0.02 | 0.01 | 0.03 | 0.28 | 0.25 | 0 | 8.16 | -2.1 | 8.42 |
| 2 | 0 | 11 | 70,958,849,724 | 0.02 | 0.01 | 0.03 | 0.28 | 0.25 | 0 | -0.65 | -0.68 | 0.94 |
| -6 | -3 | 21 | 70,958,849,724 | 0.02 | 0.01 | 0.03 | 0.28 | 0.25 | 0.12 | 0.85 | -1.3 | 1.56 |
| -5 | -7 | 20 | 70,958,849,724 | 0.02 | 0.01 | 0.03 | 0.28 | 0.25 | 0.27 | -0.95 | -1.24 | 1.58 |
| -1 | -5 | 21 | 70,958,849,724 | 0.02 | 0.01 | 0.03 | 0.28 | 0.25 | 0.19 | -1.52 | -1.3 | 2.01 |
| 0 | -5 | 18 | 70,958,849,724 | 0.02 | 0.01 | 0.03 | 0.28 | 0.25 | 0.19 | -1.84 | -1.11 | 2.16 |
| 1 | -6 | 20 | 70,958,849,724 | 0.02 | 0.01 | 0.03 | 0.28 | 0.25 | 0.23 | -2.54 | -1.24 | 2.83 |
| 4 | -6 | 22 | 70,958,849,724 | 0.02 | 0.01 | 0.03 | 0.28 | 0.25 | 0.23 | -3.52 | -1.36 | 3.78 |
| 3 | -5 | 5 | 70,958,849,724 | 0.02 | 0.01 | 0.03 | 0.28 | 0.25 | 0.19 | -2.82 | -0.31 | 2.85 |
| 0 | 0 | 0 | 70,958,849,724 | 0.02 | 0.01 | 0.03 | 0.28 | 0.25 | 0 | 0 | 0 | 0 |
| 0 | 0 | 0 | 70,958,849,724 | 0.02 | 0.01 | 0.03 | 0.28 | 0.25 | 0 | 0 | 0 | 0 |
| 0 | 0 | 0 | 70,958,849,724 | 0.02 | 0.01 | 0.03 | 0.28 | 0.25 | 0 | 0 | 0 | 0 |

**Activation 4**

| $\varepsilon_H$ (με) | $\varepsilon_V$ (με) | $\varepsilon_L$ (με) | K (Pa) | b (m) | h (m) | L (m) | H (m) | HLV (m) | Fx (N) | Fy (N) | Fz (N) | AF (N) |
|---|---|---|---|---|---|---|---|---|---|---|---|---|
| 0 | -3 | 4 | 70,958,849,724 | 0.02 | 0.01 | 0.03 | 0.28 | 0.25 | 0.12 | -1.11 | -0.25 | 1.14 |
| -1 | -1 | 3 | 70,958,849,724 | 0.02 | 0.01 | 0.03 | 0.28 | 0.25 | 0.04 | -0.04 | -0.19 | 0.19 |
| 34 | -9 | 26 | 70,958,849,724 | 0.02 | 0.01 | 0.03 | 0.28 | 0.25 | 0.35 | -14.41 | -1.61 | 14.5 |
| -2 | -12 | -4 | 70,958,849,724 | 0.02 | 0.01 | 0.03 | 0.28 | 0.25 | 0.46 | -3.77 | 0.25 | 3.81 |
| -7 | -5 | 29 | 70,958,849,724 | 0.02 | 0.01 | 0.03 | 0.28 | 0.25 | 0.19 | 0.44 | -1.79 | 1.85 |
| 9 | -8 | 33 | 70,958,849,724 | 0.02 | 0.01 | 0.03 | 0.28 | 0.25 | 0.31 | -5.89 | -2.04 | 6.24 |
| 8 | -6 | 31 | 70,958,849,724 | 0.02 | 0.01 | 0.03 | 0.28 | 0.25 | 0.23 | -4.82 | -1.91 | 5.19 |
| 7 | -5 | 31 | 70,958,849,724 | 0.02 | 0.01 | 0.03 | 0.28 | 0.25 | 0.19 | -4.13 | -1.91 | 4.55 |
| 8 | -5 | 30 | 70,958,849,724 | 0.02 | 0.01 | 0.03 | 0.28 | 0.25 | 0.19 | -4.45 | -1.85 | 4.83 |
| 8 | -5 | 29 | 70,958,849,724 | 0.02 | 0.01 | 0.03 | 0.28 | 0.25 | 0.19 | -4.45 | -1.79 | 4.8 |
| 0 | -1 | 0 | 70,958,849,724 | 0.02 | 0.01 | 0.03 | 0.28 | 0.25 | 0.04 | -0.37 | 0 | 0.37 |
| 0 | 0 | 0 | 70,958,849,724 | 0.02 | 0.01 | 0.03 | 0.28 | 0.25 | 0 | 0 | 0 | 0 |
| 0 | 0 | 0 | 70,958,849,724 | 0.02 | 0.01 | 0.03 | 0.28 | 0.25 | 0 | 0 | 0 | 0 |

**Activation 5**

| $\varepsilon_H$ (με) | $\varepsilon_V$ (με) | $\varepsilon_L$ (με) | K (Pa) | b (m) | h (m) | L (m) | H (m) | HLV (m) | Fx (N) | Fy (N) | Fz (N) | AF (N) |
|---|---|---|---|---|---|---|---|---|---|---|---|---|
| 1 | 0 | 3 | 70,958,849,724 | 0.02 | 0.01 | 0.03 | 0.28 | 0.25 | 0 | -0.33 | -0.19 | 0.38 |
| -25 | 0 | 34 | 70,958,849,724 | 0.02 | 0.01 | 0.03 | 0.28 | 0.25 | 0 | 8.16 | -2.1 | 8.42 |
| 2 | 0 | 11 | 70,958,849,724 | 0.02 | 0.01 | 0.03 | 0.28 | 0.25 | 0 | -0.65 | -0.68 | 0.94 |
| -6 | -3 | 21 | 70,958,849,724 | 0.02 | 0.01 | 0.03 | 0.28 | 0.25 | 0.12 | 0.85 | -1.3 | 1.56 |
| -5 | -7 | 20 | 70,958,849,724 | 0.02 | 0.01 | 0.03 | 0.28 | 0.25 | 0.27 | -0.95 | -1.24 | 1.58 |
| -1 | -5 | 21 | 70,958,849,724 | 0.02 | 0.01 | 0.03 | 0.28 | 0.25 | 0.19 | -1.52 | -1.3 | 2.01 |
| 0 | -5 | 18 | 70,958,849,724 | 0.02 | 0.01 | 0.03 | 0.28 | 0.25 | 0.19 | -1.84 | -1.11 | 2.16 |
| 1 | -6 | 20 | 70,958,849,724 | 0.02 | 0.01 | 0.03 | 0.28 | 0.25 | 0.23 | -2.54 | -1.24 | 2.83 |
| 4 | -6 | 22 | 70,958,849,724 | 0.02 | 0.01 | 0.03 | 0.28 | 0.25 | 0.23 | -3.52 | -1.36 | 3.78 |
| 3 | -5 | 5 | 70,958,849,724 | 0.02 | 0.01 | 0.03 | 0.28 | 0.25 | 0.19 | -2.82 | -0.31 | 2.85 |
| 0 | 0 | 0 | 70,958,849,724 | 0.02 | 0.01 | 0.03 | 0.28 | 0.25 | 0 | 0 | 0 | 0 |
| 0 | 0 | 0 | 70,958,849,724 | 0.02 | 0.01 | 0.03 | 0.28 | 0.25 | 0 | 0 | 0 | 0 |
| 0 | 0 | 0 | 70,958,849,724 | 0.02 | 0.01 | 0.03 | 0.28 | 0.25 | 0 | 0 | 0 | 0 |

# Test P5 Female User

### Activation 1

| $\varepsilon_H$ (με) | $\varepsilon_V$ (με) | $\varepsilon_L$ (με) | K (Pa) | b (m) | h (m) | L (m) | H (m) | HLV (m) | Fx (N) | Fy (N) | Fz (N) | AF (N) |
|---|---|---|---|---|---|---|---|---|---|---|---|---|
| 0 | 0 | 0 | 70,958,849,724 | 0.02 | 0.01 | 0.03 | 0.28 | 0.25 | 0 | 0 | 0 | 0 |
| -13 | 1 | -13 | 70,958,849,724 | 0.02 | 0.01 | 0.03 | 0.28 | 0.25 | -0.04 | 4.61 | 0.8 | 4.68 |
| -28 | -8 | 17 | 70,958,849,724 | 0.02 | 0.01 | 0.03 | 0.28 | 0.25 | 0.31 | 6.19 | -1.05 | 6.28 |
| 0 | 0 | 28 | 70,958,849,724 | 0.02 | 0.01 | 0.03 | 0.28 | 0.25 | 0 | 0 | -1.73 | 1.73 |
| 45 | -9 | 81 | 70,958,849,724 | 0.02 | 0.01 | 0.03 | 0.28 | 0.25 | 0.35 | -18 | -5 | 18.69 |
| 55 | -5 | 77 | 70,958,849,724 | 0.02 | 0.01 | 0.03 | 0.28 | 0.25 | 0.19 | -19.79 | -4.76 | 20.35 |
| 39 | -6 | 67 | 70,958,849,724 | 0.02 | 0.01 | 0.03 | 0.28 | 0.25 | 0.23 | -14.94 | -4.14 | 15.5 |
| 36 | -8 | 67 | 70,958,849,724 | 0.02 | 0.01 | 0.03 | 0.28 | 0.25 | 0.31 | -14.69 | -4.14 | 15.27 |
| 26 | -9 | 2 | 70,958,849,724 | 0.02 | 0.01 | 0.03 | 0.28 | 0.25 | 0.35 | -11.8 | -0.12 | 11.81 |
| 0 | 0 | 0 | 70,958,849,724 | 0.02 | 0.01 | 0.03 | 0.28 | 0.25 | 0 | 0 | 0 | 0 |
| 0 | 0 | 0 | 70,958,849,724 | 0.02 | 0.01 | 0.03 | 0.28 | 0.25 | 0 | 0 | 0 | 0 |
| 0 | 0 | 0 | 70,958,849,724 | 0.02 | 0.01 | 0.03 | 0.28 | 0.25 | 0 | 0 | 0 | 0 |
| 0 | 0 | 0 | 70,958,849,724 | 0.02 | 0.01 | 0.03 | 0.28 | 0.25 | 0 | 0 | 0 | 0 |

### Activation 2

| $\varepsilon_H$ (με) | $\varepsilon_V$ (με) | $\varepsilon_L$ (με) | K (Pa) | b (m) | h (m) | L (m) | H (m) | HLV (m) | Fx (N) | Fy (N) | Fz (N) | AF (N) |
|---|---|---|---|---|---|---|---|---|---|---|---|---|
| 0 | -2 | 24 | 70,958,849,724 | 0.02 | 0.01 | 0.03 | 0.28 | 0.25 | 0.08 | -0.74 | -1.48 | 1.66 |
| -14 | -9 | 34 | 70,958,849,724 | 0.02 | 0.01 | 0.03 | 0.28 | 0.25 | 0.35 | 1.25 | -2.1 | 2.47 |
| 12 | -11 | 37 | 70,958,849,724 | 0.02 | 0.01 | 0.03 | 0.28 | 0.25 | 0.42 | -7.97 | -2.29 | 8.3 |
| 16 | -12 | 41 | 70,958,849,724 | 0.02 | 0.01 | 0.03 | 0.28 | 0.25 | 0.46 | -9.64 | -2.53 | 9.98 |
| 13 | -11 | 40 | 70,958,849,724 | 0.02 | 0.01 | 0.03 | 0.28 | 0.25 | 0.42 | -8.3 | -2.47 | 8.67 |
| 17 | -12 | 40 | 70,958,849,724 | 0.02 | 0.01 | 0.03 | 0.28 | 0.25 | 0.46 | -9.97 | -2.47 | 10.28 |
| 15 | -11 | 37 | 70,958,849,724 | 0.02 | 0.01 | 0.03 | 0.28 | 0.25 | 0.42 | -8.95 | -2.29 | 9.25 |
| 14 | -11 | 39 | 70,958,849,724 | 0.02 | 0.01 | 0.03 | 0.28 | 0.25 | 0.42 | -8.62 | -2.41 | 8.96 |
| 11 | -9 | 26 | 70,958,849,724 | 0.02 | 0.01 | 0.03 | 0.28 | 0.25 | 0.35 | -6.91 | -1.61 | 7.1 |
| 0 | 0 | 0 | 70,958,849,724 | 0.02 | 0.01 | 0.03 | 0.28 | 0.25 | 0 | 0 | 0 | 0 |
| 0 | 0 | 0 | 70,958,849,724 | 0.02 | 0.01 | 0.03 | 0.28 | 0.25 | 0 | 0 | 0 | 0 |
| 0 | 0 | 0 | 70,958,849,724 | 0.02 | 0.01 | 0.03 | 0.28 | 0.25 | 0 | 0 | 0 | 0 |
| 0 | 0 | 0 | 70,958,849,724 | 0.02 | 0.01 | 0.03 | 0.28 | 0.25 | 0 | 0 | 0 | 0 |

### Activation 3

| $\varepsilon_H$ (με) | $\varepsilon_V$ (με) | $\varepsilon_L$ (με) | K (Pa) | b (m) | h (m) | L (m) | H (m) | HLV (m) | Fx (N) | Fy (N) | Fz (N) | AF (N) |
|---|---|---|---|---|---|---|---|---|---|---|---|---|
| 10 | -5 | 47 | 70,958,849,724 | 0.02 | 0.01 | 0.03 | 0.28 | 0.25 | 0.19 | -5.11 | -2.9 | 5.88 |
| -5 | 0 | 0 | 70,958,849,724 | 0.02 | 0.01 | 0.03 | 0.28 | 0.25 | 0 | 1.63 | 0 | 1.63 |
| 1 | 2 | 101 | 70,958,849,724 | 0.02 | 0.01 | 0.03 | 0.28 | 0.25 | -0.08 | 0.41 | -6.24 | 6.25 |
| 2 | 0 | 0 | 70,958,849,724 | 0.02 | 0.01 | 0.03 | 0.28 | 0.25 | 0 | -0.65 | 0 | 0.65 |
| -13 | 1 | 18 | 70,958,849,724 | 0.02 | 0.01 | 0.03 | 0.28 | 0.25 | -0.04 | 4.61 | -1.11 | 4.74 |
| 0 | -2 | 38 | 70,958,849,724 | 0.02 | 0.01 | 0.03 | 0.28 | 0.25 | 0.08 | -0.74 | -2.35 | 2.46 |
| 13 | 0 | 46 | 70,958,849,724 | 0.02 | 0.01 | 0.03 | 0.28 | 0.25 | 0 | -4.24 | -2.84 | 5.1 |
| 23 | -2 | 56 | 70,958,849,724 | 0.02 | 0.01 | 0.03 | 0.28 | 0.25 | 0.08 | -8.24 | -3.46 | 8.94 |
| 44 | -4 | 72 | 70,958,849,724 | 0.02 | 0.01 | 0.03 | 0.28 | 0.25 | 0.15 | -15.83 | -4.45 | 16.44 |
| 44 | -4 | 73 | 70,958,849,724 | 0.02 | 0.01 | 0.03 | 0.28 | 0.25 | 0.15 | -15.83 | -4.51 | 16.46 |
| 42 | -5 | 69 | 70,958,849,724 | 0.02 | 0.01 | 0.03 | 0.28 | 0.25 | 0.19 | -15.55 | -4.26 | 16.12 |
| 39 | -5 | 63 | 70,958,849,724 | 0.02 | 0.01 | 0.03 | 0.28 | 0.25 | 0.19 | -14.57 | -3.89 | 15.08 |
| 0 | 0 | 0 | 70,958,849,724 | 0.02 | 0.01 | 0.03 | 0.28 | 0.25 | 0 | 0 | 0 | 0 |

### Activation 4

| $\varepsilon_H$ (με) | $\varepsilon_V$ (με) | $\varepsilon_L$ (με) | K (Pa) | b (m) | h (m) | L (m) | H (m) | HLV (m) | Fx (N) | Fy (N) | Fz (N) | AF (N) |
|---|---|---|---|---|---|---|---|---|---|---|---|---|
| -1 | 0 | 0 | 70,958,849,724 | 0.02 | 0.01 | 0.03 | 0.28 | 0.25 | 0 | 0.33 | 0 | 0.33 |
| -10 | -2 | 107 | 70,958,849,724 | 0.02 | 0.01 | 0.03 | 0.28 | 0.25 | 0.08 | 2.53 | -6.61 | 7.07 |
| 4 | 0 | 0 | 70,958,849,724 | 0.02 | 0.01 | 0.03 | 0.28 | 0.25 | 0 | -1.3 | 0 | 1.3 |
| -18 | 10 | 69 | 70,958,849,724 | 0.02 | 0.01 | 0.03 | 0.28 | 0.25 | -0.39 | 9.56 | -4.26 | 10.47 |
| 30 | 0 | 51 | 70,958,849,724 | 0.02 | 0.01 | 0.03 | 0.28 | 0.25 | 0 | -9.79 | -3.15 | 10.28 |
| 27 | -3 | 53 | 70,958,849,724 | 0.02 | 0.01 | 0.03 | 0.28 | 0.25 | 0.12 | -9.91 | -3.27 | 10.44 |
| 28 | -3 | 49 | 70,958,849,724 | 0.02 | 0.01 | 0.03 | 0.28 | 0.25 | 0.12 | -10.24 | -3.03 | 10.68 |
| 29 | -3 | 49 | 70,958,849,724 | 0.02 | 0.01 | 0.03 | 0.28 | 0.25 | 0.12 | -10.57 | -3.03 | 10.99 |
| 28 | -4 | 49 | 70,958,849,724 | 0.02 | 0.01 | 0.03 | 0.28 | 0.25 | 0.15 | -10.61 | -3.03 | 11.03 |
| 33 | -4 | 53 | 70,958,849,724 | 0.02 | 0.01 | 0.03 | 0.28 | 0.25 | 0.15 | -12.24 | -3.27 | 12.67 |
| 27 | -5 | 49 | 70,958,849,724 | 0.02 | 0.01 | 0.03 | 0.28 | 0.25 | 0.19 | -10.65 | -3.03 | 11.08 |
| 0 | 0 | 0 | 70,958,849,724 | 0.02 | 0.01 | 0.03 | 0.28 | 0.25 | 0 | 0 | 0 | 0 |
| 0 | 0 | 0 | 70,958,849,724 | 0.02 | 0.01 | 0.03 | 0.28 | 0.25 | 0 | 0 | 0 | 0 |

### Activation 5

| $\varepsilon_H$ (με) | $\varepsilon_V$ (με) | $\varepsilon_L$ (με) | K (Pa) | b (m) | h (m) | L (m) | H (m) | HLV (m) | Fx (N) | Fy (N) | Fz (N) | AF (N) |
|---|---|---|---|---|---|---|---|---|---|---|---|---|
| 5 | -9 | 32 | 70,958,849,724 | 0.02 | 0.01 | 0.03 | 0.28 | 0.25 | 0.35 | -4.95 | -1.98 | 5.34 |
| -4 | 0 | 0 | 70,958,849,724 | 0.02 | 0.01 | 0.03 | 0.28 | 0.25 | 0 | 1.3 | 0 | 1.3 |
| -26 | -5 | 19 | 70,958,849,724 | 0.02 | 0.01 | 0.03 | 0.28 | 0.25 | 0.19 | 6.64 | -1.17 | 6.74 |
| 4 | -1 | 30 | 70,958,849,724 | 0.02 | 0.01 | 0.03 | 0.28 | 0.25 | 0.04 | -1.67 | -1.85 | 2.5 |
| 9 | -3 | 30 | 70,958,849,724 | 0.02 | 0.01 | 0.03 | 0.28 | 0.25 | 0.12 | -4.04 | -1.85 | 4.45 |
| 6 | -2 | 29 | 70,958,849,724 | 0.02 | 0.01 | 0.03 | 0.28 | 0.25 | 0.08 | -2.69 | -1.79 | 3.24 |
| 16 | -3 | 40 | 70,958,849,724 | 0.02 | 0.01 | 0.03 | 0.28 | 0.25 | 0.12 | -6.33 | -2.47 | 6.79 |
| 17 | -4 | 40 | 70,958,849,724 | 0.02 | 0.01 | 0.03 | 0.28 | 0.25 | 0.15 | -7.02 | -2.47 | 7.44 |
| 13 | -3 | 35 | 70,958,849,724 | 0.02 | 0.01 | 0.03 | 0.28 | 0.25 | 0.12 | -5.35 | -2.16 | 5.77 |
| 15 | -4 | 39 | 70,958,849,724 | 0.02 | 0.01 | 0.03 | 0.28 | 0.25 | 0.15 | -6.37 | -2.41 | 6.81 |
| 15 | -3 | 34 | 70,958,849,724 | 0.02 | 0.01 | 0.03 | 0.28 | 0.25 | 0.12 | -6 | -2.1 | 6.36 |
| 0 | 0 | 0 | 70,958,849,724 | 0.02 | 0.01 | 0.03 | 0.28 | 0.25 | 0 | 0 | 0 | 0 |
| 0 | 0 | 0 | 70,958,849,724 | 0.02 | 0.01 | 0.03 | 0.28 | 0.25 | 0 | 0 | 0 | 0 |

# Test P6 Male User

### Activation 1

| $\varepsilon_H$ (με) | $\varepsilon_V$ (με) | $\varepsilon_L$ (με) | K (Pa) | b (m) | h (m) | L (m) | H (m) | HLV (m) | Fx (N) | Fy (N) | Fz (N) | AF (N) |
|---|---|---|---|---|---|---|---|---|---|---|---|---|
| -3 | 3 | 20 | 70,958,849,724 | 0.02 | 0.01 | 0.03 | 0.28 | 0.25 | -0.12 | 2.08 | -1.24 | 2.43 |
| 5 | 12 | 40 | 70,958,849,724 | 0.02 | 0.01 | 0.03 | 0.28 | 0.25 | -0.46 | 2.79 | -2.47 | 3.76 |
| 0 | 6 | 22 | 70,958,849,724 | 0.02 | 0.01 | 0.03 | 0.28 | 0.25 | -0.23 | 2.21 | -1.36 | 2.61 |
| -13 | 28 | 158 | 70,958,849,724 | 0.02 | 0.01 | 0.03 | 0.28 | 0.25 | -1.08 | 14.57 | -9.76 | 17.57 |
| 0 | 2 | 3 | 70,958,849,724 | 0.02 | 0.01 | 0.03 | 0.28 | 0.25 | -0.08 | 0.74 | -0.19 | 0.76 |
| -8 | -13 | 71 | 70,958,849,724 | 0.02 | 0.01 | 0.03 | 0.28 | 0.25 | 0.5 | -2.18 | -4.39 | 4.92 |
| 68 | 6 | 136 | 70,958,849,724 | 0.02 | 0.01 | 0.03 | 0.28 | 0.25 | -0.23 | -19.97 | -8.4 | 21.67 |
| 53 | 9 | 103 | 70,958,849,724 | 0.02 | 0.01 | 0.03 | 0.28 | 0.25 | -0.35 | -13.97 | -6.36 | 15.36 |
| 50 | 1 | 98 | 70,958,849,724 | 0.02 | 0.01 | 0.03 | 0.28 | 0.25 | -0.04 | -15.94 | -6.05 | 17.05 |
| 35 | 0 | 84 | 70,958,849,724 | 0.02 | 0.01 | 0.03 | 0.28 | 0.25 | 0 | -11.42 | -5.19 | 12.54 |
| 35 | 0 | 91 | 70,958,849,724 | 0.02 | 0.01 | 0.03 | 0.28 | 0.25 | 0 | -11.42 | -5.62 | 12.73 |
| 0 | 0 | 0 | 70,958,849,724 | 0.02 | 0.01 | 0.03 | 0.28 | 0.25 | 0 | 0 | 0 | 0 |
| 0 | 0 | 0 | 70,958,849,724 | 0.02 | 0.01 | 0.03 | 0.28 | 0.25 | 0 | 0 | 0 | 0 |

### Activation 2

| $\varepsilon_H$ (με) | $\varepsilon_V$ (με) | $\varepsilon_L$ (με) | K (Pa) | b (m) | h (m) | L (m) | H (m) | HLV (m) | Fx (N) | Fy (N) | Fz (N) | AF (N) |
|---|---|---|---|---|---|---|---|---|---|---|---|---|
| 1 | -4 | 24 | 70,958,849,724 | 0.02 | 0.01 | 0.03 | 0.28 | 0.25 | 0.15 | -1.8 | -1.48 | 2.34 |
| 8 | -1 | 3 | 70,958,849,724 | 0.02 | 0.01 | 0.03 | 0.28 | 0.25 | 0.04 | -2.98 | -0.19 | 2.98 |
| 11 | -14 | 264 | 70,958,849,724 | 0.02 | 0.01 | 0.03 | 0.28 | 0.25 | 0.54 | -8.75 | -16.31 | 18.51 |
| 16 | 1 | 91 | 70,958,849,724 | 0.02 | 0.01 | 0.03 | 0.28 | 0.25 | -0.04 | -4.85 | -5.62 | 7.42 |
| 16 | -7 | 70 | 70,958,849,724 | 0.02 | 0.01 | 0.03 | 0.28 | 0.25 | 0.27 | -7.8 | -4.32 | 8.92 |
| 6 | -6 | 51 | 70,958,849,724 | 0.02 | 0.01 | 0.03 | 0.28 | 0.25 | 0.23 | -4.17 | -3.15 | 5.23 |
| 11 | -7 | 62 | 70,958,849,724 | 0.02 | 0.01 | 0.03 | 0.28 | 0.25 | 0.27 | -6.17 | -3.83 | 7.27 |
| 10 | -6 | 68 | 70,958,849,724 | 0.02 | 0.01 | 0.03 | 0.28 | 0.25 | 0.23 | -5.47 | -4.2 | 6.9 |
| 4 | -4 | 14 | 70,958,849,724 | 0.02 | 0.01 | 0.03 | 0.28 | 0.25 | 0.15 | -2.78 | -0.86 | 2.92 |
| 0 | -1 | 0 | 70,958,849,724 | 0.02 | 0.01 | 0.03 | 0.28 | 0.25 | 0.04 | -0.37 | 0 | 0.37 |
| 0 | 0 | 0 | 70,958,849,724 | 0.02 | 0.01 | 0.03 | 0.28 | 0.25 | 0 | 0 | 0 | 0 |
| 0 | 0 | 0 | 70,958,849,724 | 0.02 | 0.01 | 0.03 | 0.28 | 0.25 | 0 | 0 | 0 | 0 |
| 0 | 0 | 0 | 70,958,849,724 | 0.02 | 0.01 | 0.03 | 0.28 | 0.25 | 0 | 0 | 0 | 0 |

### Activation 3

| $\varepsilon_H$ (με) | $\varepsilon_V$ (με) | $\varepsilon_L$ (με) | K (Pa) | b (m) | h (m) | L (m) | H (m) | HLV (m) | Fx (N) | Fy (N) | Fz (N) | AF (N) |
|---|---|---|---|---|---|---|---|---|---|---|---|---|
| 0 | 0 | 3 | 70,958,849,724 | 0.02 | 0.01 | 0.03 | 0.28 | 0.25 | 0 | 0 | -0.19 | 0.19 |
| 0 | -3 | 24 | 70,958,849,724 | 0.02 | 0.01 | 0.03 | 0.28 | 0.25 | 0.12 | -1.11 | -1.48 | 1.85 |
| 20 | 0 | 57 | 70,958,849,724 | 0.02 | 0.01 | 0.03 | 0.28 | 0.25 | 0 | -6.52 | -3.52 | 7.41 |
| 0 | -1 | 23 | 70,958,849,724 | 0.02 | 0.01 | 0.03 | 0.28 | 0.25 | 0.04 | -0.37 | -1.42 | 1.47 |
| 8 | -3 | 89 | 70,958,849,724 | 0.02 | 0.01 | 0.03 | 0.28 | 0.25 | 0.12 | -3.72 | -5.5 | 6.64 |
| 11 | -9 | 75 | 70,958,849,724 | 0.02 | 0.01 | 0.03 | 0.28 | 0.25 | 0.35 | -6.91 | -4.63 | 8.32 |
| 12 | -8 | 60 | 70,958,849,724 | 0.02 | 0.01 | 0.03 | 0.28 | 0.25 | 0.31 | -6.86 | -3.71 | 7.81 |
| 6 | -7 | 53 | 70,958,849,724 | 0.02 | 0.01 | 0.03 | 0.28 | 0.25 | 0.27 | -4.54 | -3.27 | 5.6 |
| 9 | -8 | 64 | 70,958,849,724 | 0.02 | 0.01 | 0.03 | 0.28 | 0.25 | 0.31 | -5.89 | -3.95 | 7.1 |
| 14 | -9 | 49 | 70,958,849,724 | 0.02 | 0.01 | 0.03 | 0.28 | 0.25 | 0.35 | -7.89 | -3.03 | 8.45 |
| 0 | -1 | 1 | 70,958,849,724 | 0.02 | 0.01 | 0.03 | 0.28 | 0.25 | 0.04 | -0.37 | -0.06 | 0.38 |
| 0 | 0 | 0 | 70,958,849,724 | 0.02 | 0.01 | 0.03 | 0.28 | 0.25 | 0 | 0 | 0 | 0 |
| 0 | 0 | 0 | 70,958,849,724 | 0.02 | 0.01 | 0.03 | 0.28 | 0.25 | 0 | 0 | 0 | 0 |

### Activation 4

| $\varepsilon_H$ (με) | $\varepsilon_V$ (με) | $\varepsilon_L$ (με) | K (Pa) | b (m) | h (m) | L (m) | H (m) | HLV (m) | Fx (N) | Fy (N) | Fz (N) | AF (N) |
|---|---|---|---|---|---|---|---|---|---|---|---|---|
| 0 | 1 | 7 | 70,958,849,724 | 0.02 | 0.01 | 0.03 | 0.28 | 0.25 | -0.04 | 0.37 | -0.43 | 0.57 |
| 15 | 7 | 110 | 70,958,849,724 | 0.02 | 0.01 | 0.03 | 0.28 | 0.25 | -0.27 | -2.31 | -6.79 | 7.18 |
| 0 | 0 | 8 | 70,958,849,724 | 0.02 | 0.01 | 0.03 | 0.28 | 0.25 | 0 | 0 | -0.49 | 0.49 |
| 7 | -6 | 54 | 70,958,849,724 | 0.02 | 0.01 | 0.03 | 0.28 | 0.25 | 0.23 | -4.5 | -4.26 | 6.2 |
| -11 | -4 | 36 | 70,958,849,724 | 0.02 | 0.01 | 0.03 | 0.28 | 0.25 | 0.15 | 2.11 | -2.22 | 3.07 |
| -5 | -6 | 32 | 70,958,849,724 | 0.02 | 0.01 | 0.03 | 0.28 | 0.25 | 0.23 | -0.58 | -1.98 | 2.07 |
| -2 | -5 | 47 | 70,958,849,724 | 0.02 | 0.01 | 0.03 | 0.28 | 0.25 | 0.19 | -1.19 | -2.9 | 3.14 |
| -2 | -3 | 51 | 70,958,849,724 | 0.02 | 0.01 | 0.03 | 0.28 | 0.25 | 0.12 | -0.45 | -3.15 | 3.18 |
| -1 | -8 | 44 | 70,958,849,724 | 0.02 | 0.01 | 0.03 | 0.28 | 0.25 | 0.31 | -2.62 | -2.72 | 3.79 |
| 0 | 0 | 0 | 70,958,849,724 | 0.02 | 0.01 | 0.03 | 0.28 | 0.25 | 0 | 0 | 0 | 0 |
| 0 | 0 | 0 | 70,958,849,724 | 0.02 | 0.01 | 0.03 | 0.28 | 0.25 | 0 | 0 | 0 | 0 |
| 0 | 0 | 0 | 70,958,849,724 | 0.02 | 0.01 | 0.03 | 0.28 | 0.25 | 0 | 0 | 0 | 0 |

### Activation 5

| $\varepsilon_H$ (με) | $\varepsilon_V$ (με) | $\varepsilon_L$ (με) | K (Pa) | b (m) | h (m) | L (m) | H (m) | HLV (m) | Fx (N) | Fy (N) | Fz (N) | AF (N) |
|---|---|---|---|---|---|---|---|---|---|---|---|---|
| 0 | 1 | 62 | 70,958,849,724 | 0.02 | 0.01 | 0.03 | 0.28 | 0.25 | -0.04 | 0.37 | -3.83 | 3.85 |
| -11 | 3 | 0 | 70,958,849,724 | 0.02 | 0.01 | 0.03 | 0.28 | 0.25 | -0.12 | 4.69 | 0 | 4.7 |
| 19 | 2 | 77 | 70,958,849,724 | 0.02 | 0.01 | 0.03 | 0.28 | 0.25 | -0.08 | -5.46 | -4.76 | 7.24 |
| -25 | -1 | 49 | 70,958,849,724 | 0.02 | 0.01 | 0.03 | 0.28 | 0.25 | 0.04 | 7.79 | -3.03 | 8.35 |
| -1 | -3 | 51 | 70,958,849,724 | 0.02 | 0.01 | 0.03 | 0.28 | 0.25 | 0.12 | -0.78 | -3.15 | 3.25 |
| 5 | -5 | 46 | 70,958,849,724 | 0.02 | 0.01 | 0.03 | 0.28 | 0.25 | 0.19 | -3.47 | -2.84 | 4.49 |
| -1 | -2 | 48 | 70,958,849,724 | 0.02 | 0.01 | 0.03 | 0.28 | 0.25 | 0.08 | -0.41 | -2.96 | 2.99 |
| -1 | -2 | 51 | 70,958,849,724 | 0.02 | 0.01 | 0.03 | 0.28 | 0.25 | 0.08 | -0.41 | -3.15 | 3.18 |
| -3 | 0 | 40 | 70,958,849,724 | 0.02 | 0.01 | 0.03 | 0.28 | 0.25 | 0 | 0.98 | -2.47 | 2.66 |
| -10 | 5 | 2 | 70,958,849,724 | 0.02 | 0.01 | 0.03 | 0.28 | 0.25 | -0.19 | 5.11 | -0.12 | 5.11 |
| 0 | 0 | 0 | 70,958,849,724 | 0.02 | 0.01 | 0.03 | 0.28 | 0.25 | 0 | 0 | 0 | 0 |
| 0 | 0 | 0 | 70,958,849,724 | 0.02 | 0.01 | 0.03 | 0.28 | 0.25 | 0 | 0 | 0 | 0 |
| 0 | 0 | 0 | 70,958,849,724 | 0.02 | 0.01 | 0.03 | 0.28 | 0.25 | 0 | 0 | 0 | 0 |

## Test P7 Female User

### Activation 1

| εH (με) | εV (με) | εL (με) | K (Pa) | b (m) | h (m) | L (m) | H (m) | HLV (m) | Fx (N) | Fy (N) | Fz (N) | AF (N) |
|---|---|---|---|---|---|---|---|---|---|---|---|---|
| -1 | -2 | 7 | 70,958,849,724 | 0.02 | 0.01 | 0.03 | 0.28 | 0.25 | 0.08 | -0.41 | -0.43 | 0.6 |
| -2 | -3 | 0 | 70,958,849,724 | 0.02 | 0.01 | 0.03 | 0.28 | 0.25 | 0.12 | -0.45 | 0 | 0.47 |
| 1 | 0 | 1 | 70,958,849,724 | 0.02 | 0.01 | 0.03 | 0.28 | 0.25 | 0 | -0.33 | -0.06 | 0.33 |
| -9 | -11 | 9 | 70,958,849,724 | 0.02 | 0.01 | 0.03 | 0.28 | 0.25 | 0.42 | -1.12 | -0.56 | 1.32 |
| 0 | 0 | 1 | 70,958,849,724 | 0.02 | 0.01 | 0.03 | 0.28 | 0.25 | 0 | 0 | -0.06 | 0.06 |
| 6 | -16 | 75 | 70,958,849,724 | 0.02 | 0.01 | 0.03 | 0.28 | 0.25 | 0.62 | -7.86 | -4.63 | 9.14 |
| 54 | 7 | 90 | 70,958,849,724 | 0.02 | 0.01 | 0.03 | 0.28 | 0.25 | -0.27 | -15.04 | -5.56 | 16.03 |
| 47 | 8 | 88 | 70,958,849,724 | 0.02 | 0.01 | 0.03 | 0.28 | 0.25 | -0.31 | -12.38 | -5.44 | 13.53 |
| 50 | 9 | 91 | 70,958,849,724 | 0.02 | 0.01 | 0.03 | 0.28 | 0.25 | -0.35 | -12.99 | -5.62 | 14.16 |
| 44 | 8 | 85 | 70,958,849,724 | 0.02 | 0.01 | 0.03 | 0.28 | 0.25 | -0.31 | -11.41 | -5.25 | 12.56 |
| 45 | 6 | 81 | 70,958,849,724 | 0.02 | 0.01 | 0.03 | 0.28 | 0.25 | -0.23 | -12.47 | -5 | 13.44 |
| 0 | 0 | 0 | 70,958,849,724 | 0.02 | 0.01 | 0.03 | 0.28 | 0.25 | 0 | 0 | 0 | 0 |
| 0 | 0 | 0 | 70,958,849,724 | 0.02 | 0.01 | 0.03 | 0.28 | 0.25 | 0 | 0 | 0 | 0 |

### Activation 2

| εH (με) | εV (με) | εL (με) | K (Pa) | b (m) | h (m) | L (m) | H (m) | HLV (m) | Fx (N) | Fy (N) | Fz (N) | AF (N) |
|---|---|---|---|---|---|---|---|---|---|---|---|---|
| -1 | -2 | 0 | 70,958,849,724 | 0.02 | 0.01 | 0.03 | 0.28 | 0.25 | 0.08 | -0.41 | 0 | 0.42 |
| -51 | -7 | 66 | 70,958,849,724 | 0.02 | 0.01 | 0.03 | 0.28 | 0.25 | 0.27 | 14.06 | -4.08 | 14.64 |
| 114 | 19 | 181 | 70,958,849,724 | 0.02 | 0.01 | 0.03 | 0.28 | 0.25 | -0.73 | -30.19 | -11.18 | 32.2 |
| 93 | 16 | 137 | 70,958,849,724 | 0.02 | 0.01 | 0.03 | 0.28 | 0.25 | -0.62 | -24.44 | -8.46 | 25.87 |
| 82 | 17 | 125 | 70,958,849,724 | 0.02 | 0.01 | 0.03 | 0.28 | 0.25 | -0.66 | -20.48 | -7.72 | 21.9 |
| 72 | 19 | 110 | 70,958,849,724 | 0.02 | 0.01 | 0.03 | 0.28 | 0.25 | -0.73 | -16.48 | -6.79 | 17.84 |
| 74 | 19 | 118 | 70,958,849,724 | 0.02 | 0.01 | 0.03 | 0.28 | 0.25 | -0.73 | -17.14 | -7.29 | 18.64 |
| 64 | 17 | 88 | 70,958,849,724 | 0.02 | 0.01 | 0.03 | 0.28 | 0.25 | -0.66 | -14.61 | -5.44 | 15.6 |
| 0 | 0 | 0 | 70,958,849,724 | 0.02 | 0.01 | 0.03 | 0.28 | 0.25 | 0 | 0 | 0 | 0 |
| 0 | 0 | 0 | 70,958,849,724 | 0.02 | 0.01 | 0.03 | 0.28 | 0.25 | 0 | 0 | 0 | 0 |
| 0 | 0 | 0 | 70,958,849,724 | 0.02 | 0.01 | 0.03 | 0.28 | 0.25 | 0 | 0 | 0 | 0 |
| 0 | 0 | 0 | 70,958,849,724 | 0.02 | 0.01 | 0.03 | 0.28 | 0.25 | 0 | 0 | 0 | 0 |
| 0 | 0 | 0 | 70,958,849,724 | 0.02 | 0.01 | 0.03 | 0.28 | 0.25 | 0 | 0 | 0 | 0 |

### Activation 3

| εH (με) | εV (με) | εL (με) | K (Pa) | b (m) | h (m) | L (m) | H (m) | HLV (m) | Fx (N) | Fy (N) | Fz (N) | AF (N) |
|---|---|---|---|---|---|---|---|---|---|---|---|---|
| 0 | 0 | 1 | 70,958,849,724 | 0.02 | 0.01 | 0.03 | 0.28 | 0.25 | 0 | 0 | -0.06 | 0.06 |
| 0 | 0 | 22 | 70,958,849,724 | 0.02 | 0.01 | 0.03 | 0.28 | 0.25 | 0 | 0 | -1.36 | 1.36 |
| 104 | 13 | 153 | 70,958,849,724 | 0.02 | 0.01 | 0.03 | 0.28 | 0.25 | -0.5 | -29.14 | -9.45 | 30.63 |
| 46 | 12 | 88 | 70,958,849,724 | 0.02 | 0.01 | 0.03 | 0.28 | 0.25 | -0.46 | -10.58 | -5.44 | 11.91 |
| 51 | 12 | 87 | 70,958,849,724 | 0.02 | 0.01 | 0.03 | 0.28 | 0.25 | -0.46 | -12.21 | -5.37 | 13.35 |
| 50 | 12 | 85 | 70,958,849,724 | 0.02 | 0.01 | 0.03 | 0.28 | 0.25 | -0.46 | -11.89 | -5.25 | 13 |
| 54 | 10 | 92 | 70,958,849,724 | 0.02 | 0.01 | 0.03 | 0.28 | 0.25 | -0.39 | -13.93 | -5.68 | 15.05 |
| 51 | 10 | 85 | 70,958,849,724 | 0.02 | 0.01 | 0.03 | 0.28 | 0.25 | -0.39 | -12.95 | -5.25 | 13.98 |
| 46 | 6 | 67 | 70,958,849,724 | 0.02 | 0.01 | 0.03 | 0.28 | 0.25 | -0.23 | -12.8 | -4.14 | 13.45 |
| 0 | 0 | 0 | 70,958,849,724 | 0.02 | 0.01 | 0.03 | 0.28 | 0.25 | 0 | 0 | 0 | 0 |
| 0 | 0 | 0 | 70,958,849,724 | 0.02 | 0.01 | 0.03 | 0.28 | 0.25 | 0 | 0 | 0 | 0 |
| 0 | 0 | 0 | 70,958,849,724 | 0.02 | 0.01 | 0.03 | 0.28 | 0.25 | 0 | 0 | 0 | 0 |
| 0 | 0 | 0 | 70,958,849,724 | 0.02 | 0.01 | 0.03 | 0.28 | 0.25 | 0 | 0 | 0 | 0 |

### Activation 4

| εH (με) | εV (με) | εL (με) | K (Pa) | b (m) | h (m) | L (m) | H (m) | HLV (m) | Fx (N) | Fy (N) | Fz (N) | AF (N) |
|---|---|---|---|---|---|---|---|---|---|---|---|---|
| 0 | 0 | 0 | 70,958,849,724 | 0.02 | 0.01 | 0.03 | 0.28 | 0.25 | 0 | 0 | 0 | 0 |
| 1 | 1 | 7 | 70,958,849,724 | 0.02 | 0.01 | 0.03 | 0.28 | 0.25 | -0.04 | 0.04 | -0.43 | 0.44 |
| -16 | -8 | 4 | 70,958,849,724 | 0.02 | 0.01 | 0.03 | 0.28 | 0.25 | 0.31 | 2.27 | -0.25 | 2.3 |
| 89 | 25 | 133 | 70,958,849,724 | 0.02 | 0.01 | 0.03 | 0.28 | 0.25 | -0.97 | -19.82 | -8.21 | 21.47 |
| 67 | 25 | 101 | 70,958,849,724 | 0.02 | 0.01 | 0.03 | 0.28 | 0.25 | -0.97 | -12.64 | -6.24 | 14.13 |
| 60 | 22 | 94 | 70,958,849,724 | 0.02 | 0.01 | 0.03 | 0.28 | 0.25 | -0.85 | -11.46 | -5.81 | 12.88 |
| 56 | 22 | 91 | 70,958,849,724 | 0.02 | 0.01 | 0.03 | 0.28 | 0.25 | -0.85 | -10.16 | -5.62 | 11.64 |
| 58 | 21 | 95 | 70,958,849,724 | 0.02 | 0.01 | 0.03 | 0.28 | 0.25 | -0.81 | -11.18 | -5.87 | 12.65 |
| 57 | 22 | 88 | 70,958,849,724 | 0.02 | 0.01 | 0.03 | 0.28 | 0.25 | -0.85 | -10.48 | -5.44 | 11.84 |
| -1 | 0 | 0 | 70,958,849,724 | 0.02 | 0.01 | 0.03 | 0.28 | 0.25 | 0 | 0.33 | 0 | 0.33 |
| 0 | 0 | 0 | 70,958,849,724 | 0.02 | 0.01 | 0.03 | 0.28 | 0.25 | 0 | 0 | 0 | 0 |
| 0 | 0 | 0 | 70,958,849,724 | 0.02 | 0.01 | 0.03 | 0.28 | 0.25 | 0 | 0 | 0 | 0 |
| 0 | 0 | 0 | 70,958,849,724 | 0.02 | 0.01 | 0.03 | 0.28 | 0.25 | 0 | 0 | 0 | 0 |

### Activation 5

| εH (με) | εV (με) | εL (με) | K (Pa) | b (m) | h (m) | L (m) | H (m) | HLV (m) | Fx (N) | Fy (N) | Fz (N) | AF (N) |
|---|---|---|---|---|---|---|---|---|---|---|---|---|
| -1 | 0 | 0 | 70,958,849,724 | 0.02 | 0.01 | 0.03 | 0.28 | 0.25 | 0 | 0.33 | 0 | 0.33 |
| 4 | -3 | -2 | 70,958,849,724 | 0.02 | 0.01 | 0.03 | 0.28 | 0.25 | 0.12 | -2.41 | 0.12 | 2.42 |
| 98 | -17 | 175 | 70,958,849,724 | 0.02 | 0.01 | 0.03 | 0.28 | 0.25 | 0.66 | -38.24 | -10.81 | 39.74 |
| 63 | 10 | 99 | 70,958,849,724 | 0.02 | 0.01 | 0.03 | 0.28 | 0.25 | -0.39 | -16.87 | -6.11 | 17.94 |
| 54 | 6 | 99 | 70,958,849,724 | 0.02 | 0.01 | 0.03 | 0.28 | 0.25 | -0.23 | -15.41 | -6.11 | 16.58 |
| 47 | 8 | 76 | 70,958,849,724 | 0.02 | 0.01 | 0.03 | 0.28 | 0.25 | -0.31 | -12.38 | -4.69 | 13.25 |
| 38 | 8 | 85 | 70,958,849,724 | 0.02 | 0.01 | 0.03 | 0.28 | 0.25 | -0.31 | -9.45 | -5.25 | 10.81 |
| 42 | 5 | 64 | 70,958,849,724 | 0.02 | 0.01 | 0.03 | 0.28 | 0.25 | -0.19 | -11.86 | -3.95 | 12.5 |
| 32 | 0 | 35 | 70,958,849,724 | 0.02 | 0.01 | 0.03 | 0.28 | 0.25 | 0 | -10.44 | -2.16 | 10.66 |
| -1 | 0 | 0 | 70,958,849,724 | 0.02 | 0.01 | 0.03 | 0.28 | 0.25 | 0 | 0.33 | 0 | 0.33 |
| 0 | 0 | 0 | 70,958,849,724 | 0.02 | 0.01 | 0.03 | 0.28 | 0.25 | 0 | 0 | 0 | 0 |
| 0 | 0 | 0 | 70,958,849,724 | 0.02 | 0.01 | 0.03 | 0.28 | 0.25 | 0 | 0 | 0 | 0 |
| 0 | 0 | 0 | 70,958,849,724 | 0.02 | 0.01 | 0.03 | 0.28 | 0.25 | 0 | 0 | 0 | 0 |

## Test P8 Male User

### Activation 1

| εH (με) | εV (με) | εL (με) | K (Pa) | b (m) | h (m) | L (m) | H (m) | HLV (m) | Fx (N) | Fy (N) | Fz (N) | AF (N) |
|---|---|---|---|---|---|---|---|---|---|---|---|---|
| 46 | -4 | -1 | 70,958,849,724 | 0.02 | 0.01 | 0.03 | 0.28 | 0.25 | 0.15 | -16.48 | 0.06 | 16.48 |
| 9 | -9 | 11 | 70,958,849,724 | 0.02 | 0.01 | 0.03 | 0.28 | 0.25 | 0.35 | -6.25 | -0.68 | 6.3 |
| 6 | -2 | 65 | 70,958,849,724 | 0.02 | 0.01 | 0.03 | 0.28 | 0.25 | 0.08 | -2.69 | -4.01 | 4.84 |
| 129 | -5 | 167 | 70,958,849,724 | 0.02 | 0.01 | 0.03 | 0.28 | 0.25 | 0.19 | -43.93 | -10.31 | 45.12 |
| 93 | 0 | 133 | 70,958,849,724 | 0.02 | 0.01 | 0.03 | 0.28 | 0.25 | 0 | -30.34 | -8.21 | 31.43 |
| 85 | 2 | 125 | 70,958,849,724 | 0.02 | 0.01 | 0.03 | 0.28 | 0.25 | -0.08 | -26.99 | -7.72 | 28.08 |
| 87 | 3 | 120 | 70,958,849,724 | 0.02 | 0.01 | 0.03 | 0.28 | 0.25 | -0.12 | -27.28 | -7.41 | 28.27 |
| 79 | 2 | 109 | 70,958,849,724 | 0.02 | 0.01 | 0.03 | 0.28 | 0.25 | -0.08 | -25.04 | -6.73 | 25.93 |
| 21 | -2 | 0 | 70,958,849,724 | 0.02 | 0.01 | 0.03 | 0.28 | 0.25 | 0.08 | -7.59 | 0 | 7.59 |
| 0 | 0 | 0 | 70,958,849,724 | 0.02 | 0.01 | 0.03 | 0.28 | 0.25 | 0 | 0 | 0 | 0 |
| 0 | 0 | 0 | 70,958,849,724 | 0.02 | 0.01 | 0.03 | 0.28 | 0.25 | 0 | 0 | 0 | 0 |
| 0 | 0 | 0 | 70,958,849,724 | 0.02 | 0.01 | 0.03 | 0.28 | 0.25 | 0 | 0 | 0 | 0 |

### Activation 2

| εH (με) | εV (με) | εL (με) | K (Pa) | b (m) | h (m) | L (m) | H (m) | HLV (m) | Fx (N) | Fy (N) | Fz (N) | AF (N) |
|---|---|---|---|---|---|---|---|---|---|---|---|---|
| 0 | 0 | 0 | 70,958,849,724 | 0.02 | 0.01 | 0.03 | 0.28 | 0.25 | 0 | 0 | 0 | 0 |
| 2 | 0 | 133 | 70,958,849,724 | 0.02 | 0.01 | 0.03 | 0.28 | 0.25 | 0 | -0.65 | -8.21 | 8.24 |
| 60 | 3 | 88 | 70,958,849,724 | 0.02 | 0.01 | 0.03 | 0.28 | 0.25 | -0.12 | -18.47 | -5.44 | 19.25 |
| 52 | 2 | 75 | 70,958,849,724 | 0.02 | 0.01 | 0.03 | 0.28 | 0.25 | -0.08 | -16.23 | -4.63 | 16.88 |
| 47 | 1 | 70 | 70,958,849,724 | 0.02 | 0.01 | 0.03 | 0.28 | 0.25 | -0.04 | -14.96 | -4.32 | 15.58 |
| 48 | 2 | 73 | 70,958,849,724 | 0.02 | 0.01 | 0.03 | 0.28 | 0.25 | -0.08 | -14.92 | -4.51 | 15.59 |
| 48 | 2 | 73 | 70,958,849,724 | 0.02 | 0.01 | 0.03 | 0.28 | 0.25 | 0 | -15.66 | -4.51 | 16.3 |
| 38 | 2 | 63 | 70,958,849,724 | 0.02 | 0.01 | 0.03 | 0.28 | 0.25 | -0.08 | -11.66 | -3.89 | 12.29 |
| 17 | -3 | 17 | 70,958,849,724 | 0.02 | 0.01 | 0.03 | 0.28 | 0.25 | 0.12 | -6.65 | -1.05 | 6.74 |
| 0 | 0 | 0 | 70,958,849,724 | 0.02 | 0.01 | 0.03 | 0.28 | 0.25 | 0 | 0 | 0 | 0 |
| 0 | 0 | 0 | 70,958,849,724 | 0.02 | 0.01 | 0.03 | 0.28 | 0.25 | 0 | 0 | 0 | 0 |
| 0 | 0 | 0 | 70,958,849,724 | 0.02 | 0.01 | 0.03 | 0.28 | 0.25 | 0 | 0 | 0 | 0 |

### Activation 3

| εH (με) | εV (με) | εL (με) | K (Pa) | b (m) | h (m) | L (m) | H (m) | HLV (m) | Fx (N) | Fy (N) | Fz (N) | AF (N) |
|---|---|---|---|---|---|---|---|---|---|---|---|---|
| -1 | 0 | 43 | 70,958,849,724 | 0.02 | 0.01 | 0.03 | 0.28 | 0.25 | 0 | 0.33 | -2.66 | 2.68 |
| 85 | 6 | 0 | 70,958,849,724 | 0.02 | 0.01 | 0.03 | 0.28 | 0.25 | -0.23 | -25.52 | 0 | 25.52 |
| 5 | -4 | 25 | 70,958,849,724 | 0.02 | 0.01 | 0.03 | 0.28 | 0.25 | 0.15 | -3.11 | -1.54 | 3.47 |
| 36 | -6 | 100 | 70,958,849,724 | 0.02 | 0.01 | 0.03 | 0.28 | 0.25 | 0.23 | -13.96 | -5.44 | 15.26 |
| 54 | 1 | 101 | 70,958,849,724 | 0.02 | 0.01 | 0.03 | 0.28 | 0.25 | -0.04 | -17.25 | -6.24 | 18.34 |
| 48 | 2 | 78 | 70,958,849,724 | 0.02 | 0.01 | 0.03 | 0.28 | 0.25 | -0.08 | -14.92 | -4.82 | 15.68 |
| 38 | 2 | 70 | 70,958,849,724 | 0.02 | 0.01 | 0.03 | 0.28 | 0.25 | -0.08 | -11.66 | -4.32 | 12.44 |
| 41 | 2 | 80 | 70,958,849,724 | 0.02 | 0.01 | 0.03 | 0.28 | 0.25 | -0.08 | -12.64 | -4.94 | 13.57 |
| 36 | 2 | 67 | 70,958,849,724 | 0.02 | 0.01 | 0.03 | 0.28 | 0.25 | -0.08 | -11.01 | -4.14 | 11.76 |
| 37 | 2 | 65 | 70,958,849,724 | 0.02 | 0.01 | 0.03 | 0.28 | 0.25 | -0.08 | -11.33 | -4.01 | 12.02 |
| 40 | -2 | 36 | 70,958,849,724 | 0.02 | 0.01 | 0.03 | 0.28 | 0.25 | 0.08 | -13.79 | -2.22 | 13.97 |
| 0 | 0 | 0 | 70,958,849,724 | 0.02 | 0.01 | 0.03 | 0.28 | 0.25 | 0 | 0 | 0 | 0 |
| 0 | 0 | 0 | 70,958,849,724 | 0.02 | 0.01 | 0.03 | 0.28 | 0.25 | 0 | 0 | 0 | 0 |

### Activation 4

| εH (με) | εV (με) | εL (με) | K (Pa) | b (m) | h (m) | L (m) | H (m) | HLV (m) | Fx (N) | Fy (N) | Fz (N) | AF (N) |
|---|---|---|---|---|---|---|---|---|---|---|---|---|
| -5 | 2 | 6 | 70,958,849,724 | 0.02 | 0.01 | 0.03 | 0.28 | 0.25 | -0.08 | 2.37 | -0.37 | 2.4 |
| 44 | 16 | 110 | 70,958,849,724 | 0.02 | 0.01 | 0.03 | 0.28 | 0.25 | -0.62 | -8.46 | -6.79 | 10.86 |
| 0 | -1 | 4 | 70,958,849,724 | 0.02 | 0.01 | 0.03 | 0.28 | 0.25 | 0.04 | -0.37 | -0.25 | 0.45 |
| 26 | 10 | 128 | 70,958,849,724 | 0.02 | 0.01 | 0.03 | 0.28 | 0.25 | -0.39 | -4.8 | -7.91 | 9.25 |
| 78 | 4 | 111 | 70,958,849,724 | 0.02 | 0.01 | 0.03 | 0.28 | 0.25 | -0.15 | -23.97 | -6.86 | 24.93 |
| 60 | 3 | 92 | 70,958,849,724 | 0.02 | 0.01 | 0.03 | 0.28 | 0.25 | -0.12 | -18.47 | -5.68 | 19.32 |
| 50 | 2 | 76 | 70,958,849,724 | 0.02 | 0.01 | 0.03 | 0.28 | 0.25 | -0.08 | -15.57 | -4.69 | 16.27 |
| 46 | 2 | 78 | 70,958,849,724 | 0.02 | 0.01 | 0.03 | 0.28 | 0.25 | -0.08 | -14.27 | -4.82 | 15.06 |
| 53 | 0 | 85 | 70,958,849,724 | 0.02 | 0.01 | 0.03 | 0.28 | 0.25 | 0 | -17.29 | -5.25 | 18.07 |
| 55 | 2 | 87 | 70,958,849,724 | 0.02 | 0.01 | 0.03 | 0.28 | 0.25 | -0.08 | -17.21 | -5.37 | 18.03 |
| 51 | 2 | 82 | 70,958,849,724 | 0.02 | 0.01 | 0.03 | 0.28 | 0.25 | -0.08 | -15.9 | -5.06 | 16.69 |
| 49 | -3 | 51 | 70,958,849,724 | 0.02 | 0.01 | 0.03 | 0.28 | 0.25 | 0.12 | -17.09 | -3.15 | 17.38 |
| 0 | 0 | 0 | 70,958,849,724 | 0.02 | 0.01 | 0.03 | 0.28 | 0.25 | 0 | 0 | 0 | 0 |

### Activation 5

| εH (με) | εV (με) | εL (με) | K (Pa) | b (m) | h (m) | L (m) | H (m) | HLV (m) | Fx (N) | Fy (N) | Fz (N) | AF (N) |
|---|---|---|---|---|---|---|---|---|---|---|---|---|
| 0 | 0 | 0 | 70,958,849,724 | 0.02 | 0.01 | 0.03 | 0.28 | 0.25 | 0 | 0 | 0 | 0 |
| 2 | -1 | 15 | 70,958,849,724 | 0.02 | 0.01 | 0.03 | 0.28 | 0.25 | 0.04 | -1.02 | -0.93 | 1.38 |
| 137 | -1 | 151 | 70,958,849,724 | 0.02 | 0.01 | 0.03 | 0.28 | 0.25 | 0.04 | -45.06 | -9.33 | 46.02 |
| 78 | 3 | 105 | 70,958,849,724 | 0.02 | 0.01 | 0.03 | 0.28 | 0.25 | -0.12 | -24.34 | -6.49 | 25.19 |
| 76 | 5 | 103 | 70,958,849,724 | 0.02 | 0.01 | 0.03 | 0.28 | 0.25 | -0.19 | -22.95 | -6.36 | 23.82 |
| 79 | 4 | 114 | 70,958,849,724 | 0.02 | 0.01 | 0.03 | 0.28 | 0.25 | -0.15 | -24.3 | -7.04 | 25.3 |
| 82 | 4 | 113 | 70,958,849,724 | 0.02 | 0.01 | 0.03 | 0.28 | 0.25 | -0.15 | -25.28 | -6.98 | 26.22 |
| 65 | 3 | 94 | 70,958,849,724 | 0.02 | 0.01 | 0.03 | 0.28 | 0.25 | -0.12 | -20.1 | -5.81 | 20.92 |
| 73 | 1 | 100 | 70,958,849,724 | 0.02 | 0.01 | 0.03 | 0.28 | 0.25 | -0.04 | -23.45 | -6.18 | 24.25 |
| 22 | -5 | -6 | 70,958,849,724 | 0.02 | 0.01 | 0.03 | 0.28 | 0.25 | 0.19 | -9.02 | 0.37 | 9.03 |
| 0 | 0 | 0 | 70,958,849,724 | 0.02 | 0.01 | 0.03 | 0.28 | 0.25 | 0 | 0 | 0 | 0 |
| 0 | 0 | 0 | 70,958,849,724 | 0.02 | 0.01 | 0.03 | 0.28 | 0.25 | 0 | 0 | 0 | 0 |
| 0 | 0 | 0 | 70,958,849,724 | 0.02 | 0.01 | 0.03 | 0.28 | 0.25 | 0 | 0 | 0 | 0 |

**Appendix E**

Anthropometric Data.

| Test | Age | Gender | Hand Length (m) | Hand Mass (kg) | History of Thumb Pain |
|------|-----|--------|-----------------|----------------|------------------------|
| P1 | 21 | Female | 0.173 | 0.408 | Yes |
| P2 | 22 | Male | 0.187 | 0.480 | Yes |
| P3 | 20 | Female | 0.171 | 0.402 | No |
| P4 | 20 | Male | 0.180 | 0.300 | No |
| P5 | 22 | Female | 0.188 | 0.630 | No |
| P6 | 23 | Male | 0.193 | 0.510 | No |
| P7 | 21 | Female | 0.181 | 0.444 | No |
| P8 | 24 | Male | 0.197 | 0.480 | No |

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
