# Peer review of "Triaxial Load Cell for Ergonomic Risk Assessment: A Study Case of Applied Force of Thumb"

_applsci, doi:10.3390/app14103981_

Round 1

Reviewer 1 Report

Comments and Suggestions for Authors

Line 47: I do not recommend using the phrase “ergonomic risk” as ergonomic is the study of people's efficiency in their working environment. It is a tool to evaluate workplace safety related to WMSDs, it is not a cause for injury. I would recommend saying ergonomic related risk or hazards.

I recommend avoiding the first-person tense, like we. For example, in Line 169. We need to take measurements. I recommend rewrite as “Measurements were taken ….

Line 334: I recommend reporting the mean plus the standard deviation or range (even better) to show the dispersion in addition to the central tendency. 

The load cell prototype was only used to evaluate one specific task which is lighter activation by the thumb for only 5 times. To validate the study and applicability of the load cell, I recommend testing the load cell in multiple environments and/or tools and report the results. This will allow more reliable testing results and consequently allow a better discussion of the load cell limitations and applicability. The load cell limitations should be discussed more. The authors only indicated that the load cell is task/tool specific and should be redesigned based on the task/tools being evaluated.

Another issue: The load cell was tested 5 times, why that number of repetitions was selected? Why not 10 times or 15? Were these 5 times after each other or there was a rest in between the trials? There was a huge variation in the measured thumb forces, could that be explained?

Most of the references used are old, I wish they use or cite more current studies (within the last five years). 

Comments on the Quality of English Language

The quality of English writing is ok but could be improved. I highlighted some grammatical errors and typos in yellow in the attached copy of the manuscript. 

Grammatical errors and typos: I highlighted most of them in yellow to be easily identified.

Line 12: the task is completed not developed.

Line 39: conducted not implemented.

Line 47: …. Loads, which is considered …

Line 89: monitory is a noun and you need a verb, use monitor

Line 110: compared not compare. Line 111: examined not examine

Line 170: active can't be a verb. Activates

Line 187: as described not as is described.

Reviewer 2 Report

Comments and Suggestions for Authors

Lines 292-311, 342-351, in the Results section suit more in the methods section, as the details provided there still are related to the design of the device and experiment methodology, but not the results (or only some of them, like constant K). 

Figure 4 shows the measurement of thumb AF. It is not quite clear or requires more details in describing the measurement setup - was the lighter fixed to the load cell? How much of the force was measured from the subject holding the lighter without actuating switch? 

Lines 160-163: ISO11228-3 describes reference values for holding an object between the thumb and finger, which in essence is static load and compared to the dynamic task of actuating lighter with the thumb seems not quite equivalent scenarios. More justification is required about the dynamic aspect of actuation force.

Reviewer 3 Report

Comments and Suggestions for Authors

 The paper presents a prototype of a triaxial load cell based on principles of linear elasticity theory and mechanical problems of torsion, bending and axial load. The study is interesting. The goal and the logic of the paper is clear and reasonable.

1. Introduction.

  The introduction is well written, but it needs to enrich the readers with state-of-the-art works. I suggest the authors add more related references published in recent years.

2. Methods.

(1) Figure 2. The tile of figure 2(b) was left out.

(2) An analytical model is given in details. The results were compared with the parameters established by ISO 11228-3 which was shown in Table 4. The results in table 4 were not discussed in detail. Which were results from the ISO standard and which were results from the presented research?

(3) In figure 4, it can be seen that the applied force of the thumb in a task to activate a cigarette lighter was measured. The measuring results were not presented and should be compared and discussed.

3. Discussion

  Discussion was not sufficient. I suggest the authors to improve it and compare the results with other researches.

4. Conclusions.

  The conclusions could be elaborated more in detail.

 5. References

  Some of the cited references were a little old. More related references in recent years should be added. Reference 18 was not in English form. 

Round 2

Reviewer 3 Report

Comments and Suggestions for Authors

The authors have made sufficient revisions and improvements. It can be accepted. 

Author Response

Thank you for your valuable comments. No further suggestions for change have been made.